# LIMITED REFERENCE, RELIABLE GENERATION: RULE-GUIDED TABULAR DATA GENERATION WITH DUAL-GRANULARITY FILTERING

## ABSTRACT

Synthetic tabular data generation is increasingly essential in machine learning, supporting downstream applications when real-world and high-quality tabular data is insufficient. Existing tabular generation approaches, such as generative adversarial networks (GANs), diffusion models, and fine-tuned Large Language Models (LLMs), typically require sufficient reference data, limiting their effectiveness in domain-specific datasets with scarce records. While prompt-based LLMs offer flexibility without parameter tuning, they often generate distributionally drifted data with localized redundancy, leading to degradation in downstream task performance. To overcome these issues, we propose *ReFine*, a framework that (i) derives symbolic *if–then* rules from interpretable models and embeds them into prompts to explicitly guide the generation process toward the domain-specific distribution, and (ii) applies a dual-granularity filtering that suppresses over-sampling patterns and selectively refines rare but informative samples to reduce localized redundancy. Extensive experiments on various regression and classification benchmarks demonstrate that *ReFine* consistently outperforms state-of-the-art methods, achieving up to **0.36** absolute improvement in $R^2$ for regression and **7.50%** relative improvement in $F_1$ for classification tasks.

## 1 INTRODUCTION

Tabular data serves as a foundational modality in machine learning, underpinning critical applications in domains such as healthcare, finance, and scientific research (Benjelloun et al., 2020; Ghosh, 2012; Yang et al., 2024; Shankar et al., 2024). However, the collection of large-scale, high-quality tabular datasets is often hindered by strict privacy regulations and the prohibitive costs of expert-driven annotation (Voigt & Von dem Bussche, 2017; Miceli et al., 2020). Such limitations severely restrict effective model training in many tabular applications, thereby motivating the use of synthetic data generation (Hernandez et al., 2022; Shankar et al., 2024).

Previous mainstream methods for tabular data generation are based on non-LLM generative models such as VAEs (Xu et al., 2019), GANs (Zhao et al., 2021) and diffusion models (Kotelnikov et al., 2023). Within the LLM-based paradigm, fine-tuning approaches have demonstrated notable performance (Borisov et al., 2023). Both approaches share a fundamental prerequisite: access to sufficient *reference data*, which is used as the foundation for learning underlying distributions. In practice, this prerequisite is often violated in high-stakes domains, where extremely strict privacy regulations and the rarity of critical events make large-scale, high-quality tabular datasets difficult to obtain (Kovalerchuk & Vityaev, 2005; Ji et al., 2014). For example, in rare disease diagnosis, available datasets may contain only a few dozen records, a scenario commonly referred to as a *low-data regime* (Seedat et al., 2023), which poses severe challenges for data-driven modeling (Raghavan & El Gayar, 2019; Li et al., 2023; Wang et al., 2024a). Consequently, in low-data regimes where only a handful of samples are available, existing generative methods fail to capture underlying distributions and thus struggle to produce high-quality synthetic data (Bommareddy et al., 2022; Fang et al., 2024a; Zhang et al., 2024a) In contrast, prompt-based methods exploit in-context learning to synthesize data without training, offering an alternative for low-data regimes (Seedat et al., 2023; Kim et al., 2024). However, prompt-based methods face two challenges in low-data regimes:

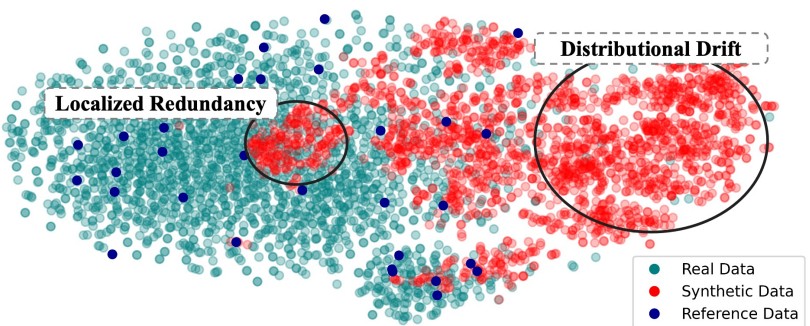

Figure 1: Two key challenges in prompt-based LLM tabular data generation in low-data regimes: (i) **Distributional Drift**: LLMs often generate biased samples by relying on spurious patterns from pretraining rather than real data distribution. (ii) **Localized Redundancy**: Synthetic samples tend to concentrate in limited regions of the feature space due to repeated use of identical prompts.

(i) **Distributional Drift**: Prompt-based methods often over-rely on LLMs' pretrained knowledge, which poorly captures dataset-specific distribution (Zhong et al., 2024). Consequently, generated samples tend to follow spurious patterns inherited from pretraining corpora rather than the true patterns of the target dataset (Sahoo et al., 2024; Stoian et al., 2024a). This mismatch leads to synthetic distributions that drifted from the *real data*. Figure 1 illustrates this issue: when conditioned on a small reference set (blue), the LLM produces *synthetic data* (red) that drift away from the manifold of *real data* (cyan). Many samples populate low-density or unsupported regions, highlighting the LLM's inability to reconstruct the target joint distribution under data scarcity.

(ii) **Localized Redundancy**: Prompt-based generation tends to overproduce high-frequency attribute combinations present in the reference data, while rare but informative patterns are rarely synthesized (Amatriain, 2024; Liu et al., 2023; Zevallos et al., 2023). This overconcentration, which we term localized redundancy, is further amplified when identical prompt templates are reused to generate data in multiple batches. As a result, synthetic samples cluster around a few dominant modes (Kim et al., 2024; Seedat et al., 2023), forming high-density regions (red) that contrast sharply with the broader and more balanced distribution of real data (cyan), as shown in Figure 1.

To systematically address the two key challenges of prompt-based tabular generation in low-data regimes, we propose **ReFine** (Rule-Guided Generation and Dual-Granularity Filtering), a framework comprising two components. To mitigate the distribution of synthetic data often drifted by LLMs in low-data regimes (Challenge i), we introduce **Rules-Guided Generation**, which extracts symbolic *if–then* formulas from interpretable tree-based models. These association rules are embedded into prompts to guide the LLM toward the distribution of the real data. To mitigate localized redundancy that persists despite prompt-based generation (Challenge ii), we propose **Dual-Granularity Filtering**. This component informs a two-level filtering process: chunk-level pruning of dominant high-density modes, and instance-level refinement to retain low-density but informative samples. Our key contributions can be summarized as follows:

- We identify two key challenges of LLM prompt-based methods in tabular data generation in low-data regimes: (i) distributional drift of the synthetic data; and (ii) localized redundancy in the synthetic data.

- To address the two challenges, we propose **ReFine**[1], a framework that constructs *association rules* to guide LLM for tabular data generation, and applies proxy-based distribution estimation with *dual-granularity* filtering to reduce localized redundancy.

- Experimental results demonstrate that **ReFine** consistently outperforms strong baselines, achieving up to **0.36** absolute gain in $R^2$ for regression and **7.5%** relative improvement in $F_1$ for classification. Comprehensive ablations further highlight the respective contributions of *Rules-Guided Generation* and *Dual-Granularity Filtering* components.

---

[1] Anonymous code: `https://anonymous.4open.science/r/ReFine-5328`

## 2 RELATED WORK

### 2.1 NON-LLM TABULAR GENERATION METHOD

**Generative Model-based generation.** Many works on tabular data synthesis relied on GANs, diffusion models, and score-based models (Hernandez et al., 2022). Among classical methods, CT-GAN (Xu et al., 2019) extends GANs to handle mixed-type variables but suffers from mode collapse and requires heavy preprocessing. TabDDPM (Kotelnikov et al., 2023) applies diffusion for continuous attributes, yet its iterative denoising is computationally costly and unstable with scarce samples. TABSYN (Zhang et al., 2024b) integrates diffusion with a VAE backbone to better support mixed-type data, but still depends on sufficient training density to avoid spurious correlations. Overall, while effective in abundant-data settings, these models degrade sharply in low-data regimes.

**Constraint-based tabular generation.** Another line of work enforces domain validity through hard logical constraints. Early methods embed linear constraints into generative models via *Constraint Layers* (Stoian et al., 2024b). More recently, *Disjunctive Refinement Layers (DRL)* extend this idea to quantifier-free real linear arithmetic (QFLRA), enabling non-convex and disjunctive feasible regions (Stoian & Giunchiglia, 2025). While effective for ensuring semantic validity, such methods face two key drawbacks: they rely on exhaustively specified domain rules, which is rarely feasible in practice, and hard constraints restrict the generation space, thereby limiting diversity.

### 2.2 LLM-BASED TABULAR GENERATION METHOD

LLMs have recently gained attention for tabular data generation, which exploit pretrained knowledge to make them well-suited for structured data tasks. Existing approaches fall into two categories: *fine-tuning methods* and *prompt-based methods*.

**Fine-Tuning Methods.** Fine-tuning methods adapt LLM parameters to tabular formats and domain constraints. For instance, GReaT (Borisov et al., 2023) fine-tunes GPT-2.5 on tabular corpora, while HARMONIC (Wang et al., 2024b) introduces instruction signals derived from nearest-neighbor relationships. Although effective with abundant reference data, these methods risk severe overfitting when reference data is small, as parameter updates dominate the limited supervision.

**Prompt-based Methods.** Prompt-based methods leverage the in-context learning ability of LLMs, enabling them to generate tabular data by conditioning on a few labeled examples embedded directly in the prompt(Ling et al., 2024). Without modifying model parameters, these methods use prompt design to guide the generation process. EPIC improves representation balance across classes by formatting grouped data and crafting class-aware prompts (Kim et al., 2024). CLLM enhances data quality in low-resource scenarios by combining prompt design with a curation step that filters samples based on model confidence and uncertainty estimates (Seedat et al., 2023). However, prompt-based methods struggle to capture logical dependencies and often suffer from distributional imbalance due to repeated use of identical prompts, limiting their effectiveness in low-data regimes.

## 3 METHODOLOGY

Prompt-based tabular generation in low-data regimes suffers from two key issues: ***Distributional Drift*** and ***Localized Redundancy***. To tackle these challenges, we unify our intuition and methodological pipeline in Figure 2. Here, the *Real Distribution* (cyan) denotes the target data manifold, while the *Original Small Dataset* (orange) provides limited supervision. To mitigate ***Distributional Drift***, we propose **(1) Rules-Guided Generation**, which extracts *Association Rules* (red) from tree-based models to explicitly capture key feature dependencies. These rules are embedded into prompts, guiding the LLM toward generation that better align with the real distribution. However, static prompting still induces ***Localized Redundancy***. To address this, we introduce **(2) Dual-Granularity Filtering**, which prunes dominant high-density chunks while retaining informative but rare samples at the instance level. Together, these two components expand the effective generation space and enhance the fidelity and diversity of the augmented dataset for downstream learning. In what follows, we first formally define the tabular generation task under low-data regimes. Then, we describe our two components for addressing the two primary challenges.

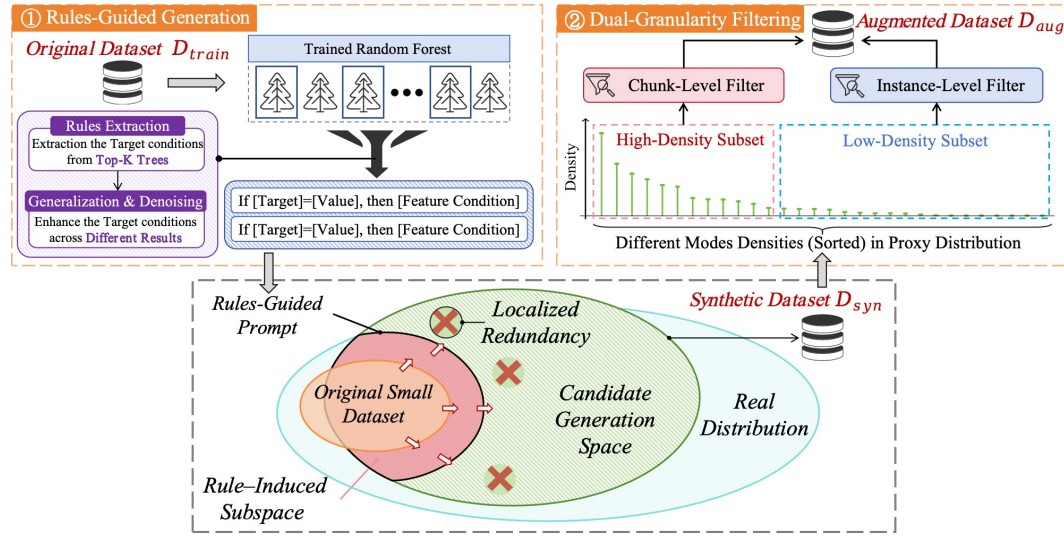

Figure 2: Overall framework of **ReFine** (*upper panel*), which consists of two components: **(1)** **Rules-Guided Generation**, which leverage rules to guide LLM generation toward the underlying *Target Distribution*; and **(2)** **Dual-Granularity Filtering**, which suppresses overrepresented patterns and preserves informative, low-density instances. The geometric view (*lower panel*) illustrates our insight—how rule-guided prompting anchors generation in a faithful *Generation Space*.

## 3.1 PROBLEM SETUP

**Definition 3.1** *Tabular Data Generation in Low-data Regimes.* Let $X \subseteq \mathbb{R}^d$ be the $d$-dimensional feature space and $Y$ the label space (discrete for classification, real-valued for regression). Let $D_{train} = \{(x_i, y_i)\}_{i=1}^N$ be a small labeled dataset independently and identically drawn from the unknown real distribution $p_R(X, Y)$, where $x_i \in X$ and $y_i \in Y$, with $N$ limited to only a few examples. The task defines a generation function $\mathcal{G}$ that maps $D_{train}$ to a synthetic dataset:

$$D_{syn} = \mathcal{G}(D_{train}) = \{(\tilde{x}_j, \tilde{y}_j)\}_{j=1}^M.$$

where $(\tilde{x}_j, \tilde{y}_j)$ is a synthetic sample, and $M$ is the number of synthetic samples, typically satisfying $M \gg N$. The task goal is to generate $D_{syn}$ such that a model trained on it achieves strong predictive performance when evaluated on a held-out real test set $D_{test} \sim p_R$.

## 3.2 COMPONENT I: RULES-GUIDED GENERATION

We mitigate *Distributional Drift* with **Rules-Guided Generation**, which extracts association rules from limited reference data $D_{\text{train}}$ as structural priors to guide generation. Unlike other modalities, tabular data often lack inherent structure and contain many irrelevant features, making it difficult for an LLM to capture feature dependencies from limited data (Fang et al., 2024b). To provide informative guidance, we define *association rules* as *if–then* formulas extracted from decision paths in tree-based models, capturing supervised dependencies between features and labels. Unlike classical association rule mining based on co-occurrence statistics, our rules reflect predictive patterns learned via training. We apply a two-stage LLM-driven procedure to extract reliable *association rules*.

**1) Rules Extraction From Top-Performing Trees.** We train a Random Forest (RF) on $D_{\text{train}}$ and rank trees based on in-sample accuracy, selecting the top-$k$ trees (e.g., $k$=3). Each selected tree provides a set of *if–else* decision rules (Kulkarni & Sinha, 2012; Azad et al., 2025; Khan et al., 2024), yielding diverse local patterns under low-data regimes.

**2) Rule Generalization and Denoising.** To construct reliable *association rules*, we apply:

*(a) Rule Generalization.* We prune and consolidate decision paths across top-performing trees, retaining only their core (i.e., high-support, low-depth) branches that reflect stable feature–label dependencies. This produces *if–then* association rules that generalize beyond individual training

instances. Treated as conditional templates with the label as premise, these rules enable inverse reasoning and steer generation toward broader yet distributionally consistent samples.

*(b) Rule Denoising.* To reduce variance introduced by symbolic extraction and decoding noise (He et al., 2024), we apply self-consistency techniques from reasoning tasks (Wang et al., 2023; Lewkowycz et al., 2022): we perform multiple generations with different seeds and retain only the most frequently occurring rules. This ensures the resulting rule set is stable and reliable.

**3) Tabular Data Generation via Rules-Guided Prompt.** Association rules obtained in Component I are converted into structured prompts encoding their *if–then* formulas. These prompts constrain the LLM's decoding space to enforce meaningful distributional patterns, yielding the synthetic dataset $D_{\text{syn}}$ that offers distribution-consistent diversity.

### 3.3 COMPONENT II: DUAL-GRANULARITY FILTERING

We introduce **Dual-Granularity Filtering** (Component II) to mitigate *localized redundancy* in the $D_{\text{syn}}$. The procedure operates at two levels: *chunk-level filtering*, which prunes over-represented modes in high-density regions, and *instance-level filtering*, which filters unreliable samples in low-density regions. Both stages are guided by a *reference model* $\mathcal{M}$ trained exclusively on $D_{\text{train}}$, ensuring that filtering remains consistent with the real data distribution.

#### 3.3.1 PROXY-BASED DENSITY ESTIMATION

We quantify local redundancy by assigning each synthetic sample to its nearest real anchor and examining the concentration of synthetic mass around anchors. Concretely, let $\{s_j\}_{j=1}^N$ denote the $N = |\mathcal{D}_{\text{train}}|$ and let DCR be the mixed-type distance from Borisov et al. (2023). This induces a discrete *proxy density* over anchors,

$$p_j \;=\; \frac{1}{|\mathcal{D}_{\text{syn}}|} \cdot \left| \left\{ x_i \in \mathcal{D}_{\text{syn}} \;\middle|\; j = \arg\min_{1 \leq k \leq n} \text{DCR}(x_i, s_k) \right\} \right|, \quad j = 1, \ldots, N. \tag{1}$$

where $x_i \in \mathcal{D}_{\text{syn}}$, $s_j \in \mathcal{D}_{\text{train}}$ and $p_j$ is the fraction of synthetic samples whose nearest neighbor in $\mathcal{D}_{\text{train}}$ is $s_j$ (thus $\sum_j p_j = 1$). We quantify overall concentration with the Gini coefficient $G(p)$; larger $G(p)$ indicates more *localized redundancy*. For targeted filtering, we split $\mathcal{D}_{\text{syn}}$ into high-density ($\mathcal{D}_{\text{high}}$) and low-density ($\mathcal{D}_{\text{low}}$) sets by $G(p)$.

#### 3.3.2 CHUNK-LEVEL FILTERING

Samples in $\mathcal{D}_{\text{high}}$ are grouped into chunks of size $S$, each denoted as $\mathcal{C}_r$. Each chunk receives a score based on the average prediction correctness across its members under $\mathcal{M}$:

$$\text{Score}(\mathcal{C}_r) = \frac{1}{|\mathcal{C}_r|} \sum_{(x_i, y_i) \in \mathcal{C}_r} \frac{1}{T} \sum_{t=1}^T \mathbb{1}\!\!\!\!/ \left( \mathbb{P}_{\mathcal{M}_t}(y_i \mid x_i) > 0.5 \right), \tag{2}$$

Chunks are then ranked by score, and only a top fraction is retained, with the *pruning ratio* adaptively scaled by the *redundancy*:

$$\text{ratio}_{\text{prune}} = A \ln(G(p)) + B, \tag{3}$$

where $A$ and $B$ are fixed constants. This adaptive schedule increases pruning strength with greater *localized redundancy*, while maintaining flexibility when the *proxy distribution* is balanced.

#### 3.3.3 INSTANCE-LEVEL FILTERING

In contrast, $\mathcal{D}_{\text{low}}$ reflects low-density regions with potentially informative but noisy samples. We filter samples using *confidence* and *uncertainty* scores derived from the same reference model $\mathcal{M}$. Thresholds are modulated by the $G(p)$:

$$\begin{aligned} \text{Conf}_{\text{thresh}} &= \mu_{\text{conf}} - G(p) \cdot \sigma_{\text{conf}}, \\ \text{Uncert}_{\text{thresh}} &= \mu_{\text{uncert}} + G(p) \cdot \sigma_{\text{uncert}}. \end{aligned} \tag{4}$$

Only samples satisfying both $\text{Conf}(x) \geq \text{Conf}_{\text{thresh}}$ and $\text{Uncert}(x) \leq \text{Uncert}_{\text{thresh}}$ are retained, ensuring filtering that adapts to dataset-specific sparsity.

### 3.3.4 Joint Tuning via Surprisal Minimization

The only tunable hyperparameter is chunk size $S$. We determine the optimal $S^*$ by minimizing model `surprisal` on $\mathcal{D}\text{train}$:

$$S^* = \arg\min_S \left[ -\frac{1}{|\mathcal{D}\text{train}|} \sum_i \log \mathbb{P}_\mathcal{M}(y_i \mid x_i) \right]. \tag{5}$$

## 4 Experiment

In this section, we evaluate **_ReFine_** on downstream tasks and analyze how its two components address the key challenges:

1. **Mitigating Distributional Drift**: *How do rules enhance data quality?* Section 4.3 shows that rule-guided generation achieves rule-compliance rates consistent with real data, thereby aligning synthetic samples more faithfully with the target distribution.

2. **Reducing Localized Redundancy**: *How does filtering balance the distribution?* Section 4.4 shows that a reliable redundancy metric together with dual-granularity filtering prevents overconcentration, leading to more useful synthetic datasets.

### 4.1 Experiment Settings

**Datasets:** To avoid potential *data contamination*—where strong performance on popular benchmarks such as *Adult*, *Heart*, and *Housing* may arise from LLM memorization rather than genuine generalization (Xu et al., 2024a; Ronval et al., 2025), we use `tabmemcheck` (Bordt et al., 2024), a recently proposed tool for detecting memorization in tabular data. Specifically, we apply two of its tests—the *Feature Names* and *Header Test*—to eight candidate datasets. Based on these tests, only *adult* and *heart* are classified as seen datasets; the remaining six show no evidence of memorization and are classified as unseen ; full results are in Appendix B.

**Baselines:** *(1) Non-LLM baselines:* (i)CTGAN (Xu et al., 2019), a GAN-based model designed for tabular data generation; and (ii) TABSYN (Zhang et al., 2024b), a score-based generative model that achieves strong performance in sufficient-data settings. *(2) LLM-based baselines:* (i) GREAT (Borisov et al., 2023), which fine-tunes a pretrained GPT 2.5 for tabular data generation; (ii) EPIC (Kim et al., 2024), a prompting-based method that automates dataset construction through instruction-driven generation; and (iii) CLLM (Seedat et al., 2023), which enhances LLM-generated data via instance-level curation. *(3) Constraint-based baseline:* DRL (Stoian & Giunchiglia, 2025), a constraint-driven generator that synthesizes data by solving logical constraints.

**Experimental Setup:** We evaluate all models under low-data regimes with $N \in \{30, 60, 90, 120\}$. For each dataset and value of $N$, we randomly sample 10 training splits, while the remaining data serve as test sets. Evaluation follows the Machine Learning Efficiency (MLE) setting: each method generates 1,000 synthetic samples per split, on which we train an XGBoost (Chen & Guestrin, 2016). Performance is reported as average **F1 score** (classification) or **$R^2$** (regression) over all splits. Further implementation details are provided in Appendix C.

### 4.2 Main Results

Table 10 reports the downstream performance of **ReFine** and representative baselines under four low-data regimes ($N \in \{30, 60, 90, 120\}$). **ReFine** achieves the strong results across all regimes, which improves $R^2$ by as much as **0.36** and $F_1$ by up to **7.5%** relative to the strongest prior method (CLLM) on unseen datasets, demonstrating robust generalization. The full **ReFine** framework (I+II) outperforms its individual components, achieving higher average ranks across benchmarks. This suggests the integration of both components contributes positively to overall effectiveness. As the amount of training data ($N$) increases, distribution-modeling capacity of non-LLM baselines, especially TabSyn, becomes more evident—they narrow the gap and occasionally approach LLM methods. This is consistent with their need for denser reference data. At the same time, DRL shows nearly flat performance across $N$, since it enforces rules as hard constraints during generation. While this guarantees strict rule adherence, it severely limits the diversity of generated samples

Table 1: Main results on benchmark datasets. We report $F_1$ **score** for classification tasks and $\mathbf{R^2}$ for regression tasks. Unseen datasets (i.e., not included in the LLM's memory for LLM-based Generator) are highlighted in blue, and seen datasets are highlighted in red. The best result in each row is shown in **bold**, and the second-best result is underlined.

| Original Data | | Non-LLM Methods | | | LLM-Based Methods | | | | | |
|---|---|---|---|---|---|---|---|---|---|---|
| Datasets | Real | CTGAN | DRL | TABSYN | GREAT | EPIC | CLLM | I \ II | II \ I | I+II |
| Disease (N=30) | 93.68 | 44.58 | 39.49 | 54.79 | 46.54 | 32.01 | 61.89 | 59.61 | 64.44 | **70.22** |
| Game (N=30) | 86.0 | 32.03 | 33.54 | 44.13 | 53.65 | 13.93 | 54.12 | 56.44 | 45.0 | **59.13** |
| Apple (N=30) | 86.60 | 47.21 | 51.33 | 32.32 | 58.91 | 56.10 | 58.18 | 57.80 | 58.66 | **59.43** |
| GPA (N=30) | 47.29 | 15.76 | 14.46 | 19.22 | 31.57 | 32.03 | 40.17 | 41.21 | 35.88 | **43.14** |
| Student (N=30) | 0.67 | -0.91 | -0.01 | 0.14 | 0.21 | 0.35 | -0.11 | 0.37 | 0.08 | **0.38** |
| Farm (N=30) | -0.04 | -0.51 | **-0.07** | -0.38 | -1.03 | -0.68 | -0.29 | -0.44 | -0.23 | -0.30 |
| Adult (N=30) | 76.92 | 50.47 | 46.59 | 62.83 | 67.45 | 61.94 | 73.11 | 73.04 | 73.15 | **73.91** |
| Heart (N=30) | 86.71 | 48.16 | 38.66 | 79.40 | 80.74 | **81.20** | 80.47 | 75.17 | 80.00 | 80.14 |
| Disease (N=60) | 93.68 | 30.03 | 38.25 | 65.34 | 52.80 | 44.05 | 66.86 | **75.65** | 62.58 | 72.41 |
| Game (N=60) | 86.0 | 31.83 | 30.10 | 61.10 | 54.76 | 13.16 | 67.54 | 61.39 | 59.81 | **70.87** |
| Apple (N=60) | 86.60 | 40.40 | 49.23 | 27.07 | 60.33 | 67.92 | 68.60 | 58.71 | **69.92** | 66.70 |
| GPA (N=60) | 47.29 | 18.11 | 15.64 | 28.48 | 35.60 | 32.19 | 33.17 | 44.34 | 21.57 | **44.58** |
| Student (N=60) | 0.67 | -0.07 | -0.04 | -0.14 | 0.02 | -0.63 | -0.48 | 0.27 | -0.15 | **0.34** |
| Farm (N=60) | -0.04 | -0.22 | -0.16 | -0.17 | -0.16 | -0.21 | -0.30 | -0.21 | **-0.12** | -0.15 |
| Adult (N=60) | 76.92 | 47.55 | 49.02 | 63.58 | 69.70 | 62.78 | **73.94** | 73.28 | 72.87 | 71.48 |
| Heart (N=60) | 86.71 | 49.77 | 38.14 | 81.35 | 80.64 | 77.55 | 80.33 | **81.37** | 78.15 | 80.36 |
| Disease (N=90) | 93.68 | 47.18 | 39.04 | 69.90 | 65.32 | 30.03 | 74.04 | 69.74 | 65.47 | **76.45** |
| Game (N=90) | 86.0 | 33.67 | 27.81 | **65.93** | 61.17 | 13.16 | 59.97 | 59.44 | 41.77 | 62.17 |
| Apple (N=90) | 86.60 | 43.08 | 38.67 | 20.69 | 64.23 | 73.67 | 73.43 | 65.23 | 72.78 | **74.88** |
| GPA (N=90) | 47.29 | 14.91 | 12.24 | 34.51 | 36.19 | 16.28 | 41.79 | 47.21 | 31.56 | **48.89** |
| Student (N=90) | 0.67 | -0.02 | -0.03 | -0.12 | 0.22 | 0.32 | -0.69 | 0.34 | -0.27 | **0.47** |
| Farm (N=90) | -0.04 | -0.34 | **-0.05** | -0.15 | -0.98 | -0.16 | -0.16 | -0.15 | -0.14 | -0.13 |
| Adult (N=90) | 76.92 | 41.67 | 47.47 | 69.77 | 71.52 | 66.73 | 74.11 | **78.45** | 73.75 | 74.10 |
| Heart (N=90) | 86.71 | 42.04 | 39.03 | 81.43 | 81.05 | 77.65 | 81.23 | **82.44** | 78.74 | 80.00 |
| Disease (N=120) | 93.68 | 45.30 | 73.92 | 62.37 | 55.70 | 55.37 | 78.30 | 78.97 | 61.82 | **81.96** |
| Game (N=120) | 86.0 | 26.95 | 31.04 | 63.50 | **66.32** | 48.99 | 61.32 | 45.48 | 54.76 | 61.67 |
| Apple (N=120) | 86.60 | 39.83 | 34.29 | **80.54** | 60.91 | 79.16 | 70.66 | 70.62 | 68.53 | 74.96 |
| GPA (N=120) | 47.29 | 16.58 | 10.63 | 38.57 | **51.29** | 46.61 | 45.20 | 40.42 | 46.95 | 47.77 |
| Student (N=120) | 0.67 | 0 | 0.01 | **0.41** | 0.23 | 0.29 | 0.22 | 0.21 | 0.17 | 0.29 |
| Farm (N=120) | -0.04 | -0.05 | **-0.01** | -0.22 | -0.18 | -0.30 | -0.23 | -0.10 | -0.29 | -0.07 |
| Adult (N=120) | 76.92 | 48.12 | 39.22 | 68.87 | 72.58 | 68.35 | 71.73 | **73.36** | 73.19 | 72.93 |
| Heart (N=120) | 86.71 | 46.29 | 38.29 | 83.80 | 84.17 | 71.33 | 75.24 | **84.97** | 75.02 | 75.93 |

and prevents DRL from exploiting richer evidence when more real data become available. By contrast, performance varies significantly among LLM-based methods. EPIC and GREAT lack explicit generation guidance, which limits their ability to enforce meaningful structure in synthetic data. The relative advantage of *ReFine* becomes smaller; in some cases (e.g., GPA), CLLM achieves a slightly higher $R^2$. This suggests that rules, while highly beneficial under data scarcity, may add mild constraints on numeric variability once sufficient reference data is available. Some LLM-based methods perform well on seen datasets but degrade notably on unseen datasets . This discrepancy aligns with the *data contamination risk* highlighted in our datasets selection—performance gains may in part reflect memorization from pretraining rather than true generalization. In contrast, *ReFine* maintains stable performance across both, validating its capability for genuine generalization. **Finally, we verify these trends beyond the main settings.** Supplementary experiments extend $N$ to $\{160, 200, 300, 500\}$ and include additional downstream evaluators (Logistic/Linear Regression and pretrained TabPFN). Across all settings and evaluators, **ReFine** remains either the best or second-best method (Tables 10–13 in Appendix F), reinforcing that its improvements are not evaluator-specific and persist when more data or alternative inductive biases are introduced.

## 4.3 MITIGATING DISTRIBUTIONAL DRIFT

*How do rules enhance data quality?* To assess the extent to which generated samples adhere to the extracted *association rules*, we define the *Rule Compliance Rate* (RCR). Given a dataset $S$ and a set of association rules $\mathcal{R}$, RCR is the percentage of samples in $S$ that satisfy all rules in $\mathcal{R}$. We compare *rule-guided* and *prompt-only* generation across three representative datasets and four low-data regimes ($N=30/60/90/120$), evaluating both rule adherence (RCR) and downstream utility. Table 2 shows that under extreme low-data regimes ($N=30$), rule-guided generation signif-

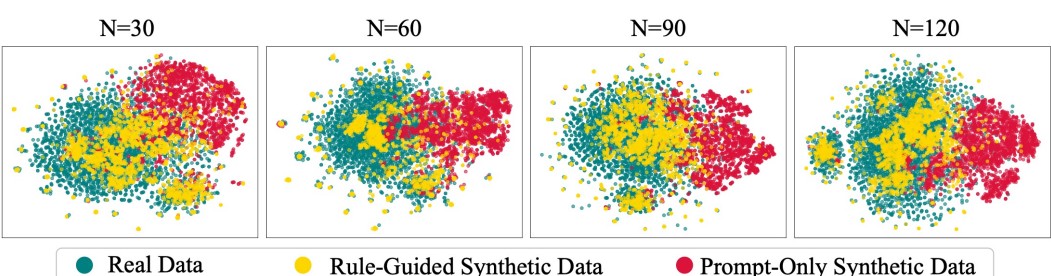

| N=30 | N=60 | N=90 | N=120 |

● Real Data   ● Rule-Guided Synthetic Data   ● Prompt-Only Synthetic Data

Figure 3: t-SNE visualization of *Real Data*, *Rule-Guided* synthetic data, and *Prompt-only* synthetic data on the *Disease* dataset across varying training sizes ($N$=30/60/90/120).

icantly compensates for the lack of distributional evidence: compared to prompt-only generation, it produces samples with both higher rule adherence and stronger downstream utility (MLE). As $N$ increases ($N$=60/120), the difference becomes most evident in *distributional alignment*. For instance, in the Disease dataset, prompt-only data attains superficially higher RCR but deviates from the real distribution and yields much lower utility. By contrast, rule-guided data achieves RCR values closer to real data and substantially improves utility. This indicates that high but distorted RCR is not reliable, and that the true advantage of rules lies in correcting such bias and aligning the synthetic distribution with the target one. The t-SNE visualizations (Figure 3) further support this finding: rule-guided samples form distributions that remain close to the real manifold, whereas prompt-only samples drift into unsupported regions. Overall, rules effectively mitigate *distributional drift* and enhance the utility of synthetic data across regimes.

We conduct further studies on *Component I* to assess both robustness and design effectiveness (Appendix D). Results show that while performance remains stable across different top-$k$ values, structured rule formats (i.e., "if-then") and self-consistency denoising yield clear improvements, validating the effectiveness of our design.

Table 2: Comparison of *rule-guided* data vs. *prompt-only* data across three datasets and varying training sizes ($N$=30/60/90/120). The better results are in bold.

|  | $D_{\text{test}}$ RCR(%) | $D_{\text{train}}$ RCR(%) | Rule Guided $D_{\text{syn}}$ RCR(%) | Rule Guided $D_{\text{syn}}$ MLE | Prompt-only $D_{\text{syn}}$ RCR(%) | Prompt-only $D_{\text{syn}}$ MLE |
|---|---|---|---|---|---|---|
| Disease (N=30) | 21.4 | 33.3 | 16.3 | **59.61** | 12.2 | 48.70 |
| GPA (N=30) | 64.5 | 70.0 | 56.7 | **41.21** | 33.8 | 26.85 |
| Student (N=30) | 33.9 | 36.7 | 46.4 | **0.37** | 15.9 | -0.11 |
| Disease (N=60) | 25.5 | 31.7 | 30.9 | **75.65** | 71.1 | 54.32 |
| GPA (N=60) | 51.5 | 61.7 | 49.1 | **44.34** | 22.5 | 15.44 |
| Student (N=60) | 80.2 | 13.3 | 16.1 | **0.27** | 15.6 | -0.49 |
| Disease (N=90) | 31.3 | 44.4 | 41.6 | **69.74** | 66.8 | 68.85 |
| GPA (N=90) | 61.1 | 51.1 | 52.7 | **47.21** | 26.4 | 27.07 |
| Student (N=90) | 53.2 | 46.7 | 55.4 | **0.34** | 26.8 | -0.70 |
| Disease (N=120) | 28.7 | 38.3 | 30.7 | **76.56** | 52.8 | 55.52 |
| GPA (N=120) | 52.6 | 53.3 | 40.8 | 40.42 | 31.6 | **44.13** |
| Student (N=120) | 57.7 | 54.2 | 67.7 | **0.21** | 17.2 | -0.46 |

## 4.4 REDUCING LOCALIZED REDUNDANCY

*How does filtering balance the distribution?* To answer this, we compare different *redundancy* metrics and granularity settings and evaluate their impact on downstream utility (MLE) across datasets and data regimes. Table 3 reveals two key insights. (i) Not all redundancy metrics are equally effective. Although both Gini and entropy cab measure redundancy, Gini consistently yields higher MLE across datasets and training sizes ($N$), particularly under data-scarce conditions. This indicates that redundancy in prompt-generated data is primarily driven by a small number of highly repeated patterns, rather than widespread noise. Gini better captures this structure by emphasizing inequality, whereas entropy averages over all regions, giving undue weight to rare, possibly noisy instances and thereby underestimating redundancy. These results emphasize dominant concentrations (e.g. Gini) are better suited to detect and suppress it. (ii) Redundancy in LLM-generated data manifests at multiple scales—both within individual samples and across batch-level concentrations. As shown in

Table 3: Evaluation of different redundancy metrics (Gini vs. Entropy) and filtering granularities (Instance-only, Chunk-only, Dual) within *Component II*.

| | Redundancy Metric | | | | Different Granularity | | |
| --- | --- | --- | --- | --- | --- | --- | --- |
| | Gini | | Entropy | | Instance | Chunk | Dual |
| | Value | MLE | Value | MLE | Only | Only | Granularity |
| Disease (n=30) | 0.40 | **70.22** | 0.13 | 68.87 | 69.29 | 63.88 | **70.22** |
| GPA (n=30) | 0.23 | **43.14** | 0.06 | 39.88 | 39.39 | 41.02 | **43.14** |
| Student (n=30) | 0.58 | **0.38** | 0.31 | 0.36 | **0.39** | 0.35 | 0.38 |
| Disease (n=60) | 0.68 | **72.41** | 0.30 | 72.03 | 71.77 | 68.81 | **72.41** |
| GPA (n=60) | 0.23 | **44.58** | 0.03 | 38.53 | 43.25 | 45.04 | **44.58** |
| Student (n=60) | 0.61 | **0.34** | 0.20 | 0.31 | 0.32 | 0.19 | **0.34** |
| Disease (n=90) | 0.40 | 72.86 | 0.21 | **76.45** | 71.65 | 71.97 | **76.45** |
| GPA (n=90) | 0.25 | **48.89** | 0.02 | 46.91 | 47.62 | 46.55 | **48.89** |
| Student (n=90) | 0.58 | **0.47** | 0.20 | 0.43 | 0.45 | 0.38 | **0.47** |
| Disease (n=120) | 0.35 | **81.96** | 0.14 | 78.97 | 80.48 | 69.74 | **81.96** |
| GPA (n=120) | 0.29 | **47.77** | 0.08 | 45.12 | 42.99 | 40.77 | **47.77** |
| Student (n=120) | 0.31 | **0.29** | 0.09 | 0.33 | 0.05 | 0.45 | **0.29** |

Table 3, dual-granularity filtering consistently outperforms instance-only and chunk-only strategies across datasets and training sizes, confirming the necessity of addressing both levels. Instance-level filtering effectively removes noisy outliers but fails to capture global distributional imbalances. In contrast, chunk-level filtering suppresses dominant high-density modes but may retain *localized redundancy*. By integrating both scales signals, dual-granularity filtering achieves more balanced coverage and higher downstream utility. This effect is visually illustrated in Figure 4: chunk-level filtering flattens overrepresented modes, while instance-level filtering enriches the long-tail regions—together restoring a more uniform and informative synthetic distribution.

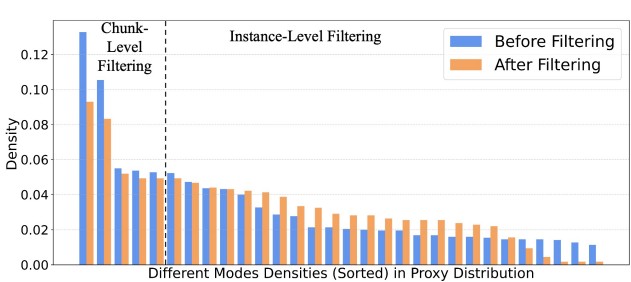

Figure 4: Proxy modes distribution before and after *dual-granularity filtering*.

We further validate the robustness of *Component II* in Appendix E. The Gini-driven pruning function peaks at intermediate retention, striking a balance between diversity and redundancy removal. Notably, Gini values stabilize with as few as 1,000 samples, confirming its suitability as a stable redundancy signal across generations.

## 5 CONCLUSION

We present **ReFine**, a two-component framework that addresses fundamental limitations of prompt-based LLM tabular generation in low-data regimes. Our approach integrates symbolic *if-then* rules derived from interpretable models to enforce domain-specific feature distribution, while employing dual-granularity filtering to mitigate localized redundancy inherent in batch generation processes. The framework demonstrates consistent improvements across diverse benchmarks, achieving up to 0.36 absolute gain in $R^2$ and 7.5% relative improvement in $F_1$ over existing methods. Component-wise analysis reveals that rule-guided generation effectively captures dataset-specific dependencies often overlooked by pre-trained models, while dual-granularity filtering successfully rebalances synthetic distributions through coordinated chunk-level and instance-level selection strategies. Nevertheless, the current implementation of chunk-level retention utilizes empirically derived logarithmic scaling, potentially limiting generalizability under extreme distributional scenarios. Future research directions include exploring adaptive retention functions and establishing theoretical foundations for improved cross-domain robustness and generalization. We leave this as an avenue for future work.

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

CONTENTS OF APPENDIX

## A   USE OF LARGE LANGUAGE MODELS (LLMS)

In preparing this manuscript, we employed a large language model (LLM) to assist with polishing the writing. Specifically, the LLM was used to improve clarity, grammar, and readability of the text. All substantive ideas, analyses, and conclusions are solely those of the authors.

# B DATASET CONTAMINATION TEST

## B.1 TABMEMCHECK

To rigorously evaluate the generalization capabilities of prompt-based tabular data generation, we assess potential training set contamination in popular benchmark datasets. Following concerns raised in recent studies (Xu et al., 2024a; Ronval et al., 2025), we employ tabmemcheck (Bordt et al., 2024), a diagnostic tool for detecting dataset memorization in LLMs. Specifically, we use two tests:

- Feature Names Test: Given a few sample rows, the LLM is prompted to infer corresponding *Column Names*.
- Header Test: The LLM is prompted to reconstruct the header (column names) and the first *few data rows* of the dataset in CSV format.

## B.2 EXAMPLE: HEART DATASET (GPT-4O)

As shown in Figure 5, GPT-4o accurately recovers the feature names and generates realistic rows for the heart dataset, despite limited prompting. Fields such as ST_Slope and RestingECG are generated verbatim, indicating memorization.

This demonstrates that **strong LLMs may replicate datasets**, compromising fair evaluation.

```
Dataset: heart.csv
Feature Names Test: Age, Sex, ChestPainType, RestingBP, Cholesterol, FastingBS, RestingECG, MaxHR, ExerciseAngina, Oldpeak, ST_Slope, HeartDisease
Legend:  Prompt Correct Incorrect Missing
Dataset: heart.csv
Header Test: Age,Sex,ChestPainType,RestingBP,Cholesterol,FastingBS,RestingECG,MaxHR,ExerciseAngina,Oldpeak,ST_Slope,HeartDisease
40,M,ATA,140,289,0,Normal,172,N,0,Up,0
49,F,NAP,160,180,0,Normal,156,N,1,Flat,1
37,M,ATA,130,283,0,ST,98,N,0,Up,0
48,F,ASY,138,214,0,Normal,108,Y,1.5,Flat,1
54,M,ATA,150,195,0,Normal,122,N,0,Up,0
39,M,NAP,120,339,0,Normal,170,N,0,Up,0
45,F,ATA,130,237,0,Normal,170,N,0,Up,0
29,M,ATA,140,280,0,Normal,170,N,0,Up,0
50,F,NAP,120,219,0,Normal,158,N,1.0,FlUp,0
```

Figure 5: **GPT-4o reproduces the heart dataset.** Field names and values are copied precisely, indicating contamination.

## B.3 EXAMPLE: ADULT DATASET (QWEN2.5-32B)

In contrast, Qwen2.5-32b fails to regenerate exact feature names but instead **identifies the dataset by name and describes its contents and use cases** (as shown in Figure 6). This demonstrates semantic-level exposure rather than strict memorization.

Such dataset-wide familiarity still violates the assumption of data independence and motivates filtering benchmarks during evaluation.

## B.4 IMPLICATIONS AND DATASET FILTERING

These examples reveal that both closed-source (GPT-4o) and open-source (Qwen2.5) models can memorize benchmarks—either by reproducing format or semantically recognizing dataset identity. Table 4 summarizes model accuracy across these tests. High scores on these tasks suggest that the

```
Info: Removed the few-shot dataset adult-train.csv because it is similar to the dataset being tested.
Dataset: adult.csv
Feature Names Test: age, workclass, fnlwgt, It seems like the list for the "adult" dataset was cut off. The Adult dataset is commonly used in machine learning tasks and typically includes information, related to income
 prediction based on census data. Below is a more complete list of feature names for the Adult dataset, which usually consists of 15 features (excluding the target variable 'income'):
Legend:  Prompt Correct Incorrect Missing
Info: Removed the few-shot dataset adult-train.csv because it is similar to the dataset being tested.
Dataset: adult.csv
Header Test: age,workclass,fnlwgt,education,education-num,marital-status,occupation,relationship,race,sex,capital-gain,capital-loss,hours-per-week,country,salary
39, State-gov,77516, Bachelors,13, Never-married, Adm-clerical, Not-in-family, White, Male,2174,0,40, United-States, <=50K
50, Self-emp-not-inc,83311, Bachelors,13, Married-civ-spouse, Exec-managerial, Husband, White, Male,0,0,13, United-States, <=50K
38, Private,215646, HS-grad,9, Divorced, Handlers-cleaners, Not-in-family, White, Male,0,0,40, United-States, <=50K
27, Private,338409, Masters,14, Married-civ-spouse, HaProf-specianlty, HusWife, White, Female,0,0,40, UnitedCubates, <>50K
32, Private,148797, Assoc-acdm,12, Married-civ-spouse, Tech-ssupporty, Wife, Black, Female,0,0,40, United-States, <=50K
34, Private,160187, Some-college,10, Married-civ-spouse, Sales, Wife, Asian-Pac-Islander, Female,0,0,40, Inited-States, <=50K
22, PSelf-e,mp-inc,231408, Bachelors,13, Never-married, QProf-specicalty, Own-cfhild, White, FeMale,0,0,45, United-States, <>50K
37, Self-ePrivat-ine,244829, Masters,14, Married-civ-spouse, Exec-managerial, Husband, Whiite, Male,0,0,80, United-States, >50K
```

Figure 6: **Qwen2.5-32b recognizes the adult dataset.** While exact headers are missing, the model describes the dataset and its usage.

Table 4: **Feature Names and Header Test** Results. **Feature Names Test**: The LLM infers column names from sample rows. **Header Test**: The LLM completes the header and first few rows of the CSV. Percentages indicate accuracy for each dataset and LLM.

| | GPT-4o-0806 | | GPT-3.5-turbo-1106 | | Qwen2.5-32b-Instruct | | Qwen2.5-14b-Instruct | |
|---|---|---|---|---|---|---|---|---|
| | Feature Names | Header | Feature Names | Header | Feature Names | Header | Feature Names | Header |
| Other Datasets | 0% | 0% | 0% | 0% | 0% | 0% | 0% | 0% |
| Adult | 86.67% | 75.0% | 86.67% | 100.0% | 0%∗ | 25.0% | 0%∗ | 80.0% |
| Heart | 100.0% | 55.56% | 25% | 22.22% | 0% | 11.11% | 0% | 12.50% |

∗Although no correct column names were produced, the LLM identified "*the Adult dataset from the UCI Machine Learning Repository*"

LLM has memorized structural or content-level information about the dataset, potentially inflating downstream performance.

## C   EXPERIMENTAL SETUP DETAILS

### C.1   OUR METHOD

**Model Configuration.**  We use GPT-3.5-Turbo-1106 as the backend model for both association rule extraction and data generation. Each baseline is configured to generate roughly 2,000 synthetic samples per dataset.

**Rule-Guided Generation.**  We set $k = 3$ for selecting top-performing trees in Random Forests, and apply self-consistency by aggregating 5 independently sampled generations for rule extraction.

**Dual-Granularity Curation.**  We use XGBoost as the reference model $\mathcal{M}$ for evaluating chunk-level informativeness, and search chunk sizes $S \in \{20, 25, \ldots, 60\}$ during tuning.

**Evaluation.**  For each run, we train XGBoost on 1,000 synthetic samples under 10 random seeds and evaluate on the corresponding real dataset. Classification tasks are measured with F1 score; regression tasks with $R^2$.

### C.2   HYPERPARAMETER SETTINGS FOR BASELINES

For **non-LLM based method**, we follow the standard training configurations reported in prior work. The detailed hyperparameter settings are summarized in Table 5.

For **LLM-based methods**, we standardize the generation temperature across all models. We use a fixed sampling temperature of 0.9 for all LLM prompt-based approaches, including baselines such as CLLM and EPIC as well as our method. Unless otherwise specified, all LLM generations use this temperature. The GReaT's hyperparameter settings are summarized in Table 5.

Table 5: Hyper-parameter settings for non-llm based baseline methods.

| Method / Phase | Epochs | Batch size | Optimizer | Initial LR | Additional settings |
|---|---|---|---|---|---|
| CTGAN | 200 | 24 | Adam | $2 \times 10^{-4}$ | — |
| GREAT | 400 | 16 | Adam | $2 \times 10^{-4}$ | |
| DRL (CTGAN-based) | 150 | 24 | Adam | $2 \times 10^{-4}$ | — |
| TabSyn – VAE phase | 3000 | 4096 | Adam | $10^{-3}$ | — |
| TabSyn – DDPM phase | up to 10001[2] | 4096 | Adam | $10^{-3}$ | LR scheduler[3] |

---

[2]Early stopping: if current loss $\geq$ best loss for 500 consecutive epochs.
[3]`ReduceLROnPlateau` (PyTorch) applied during the DDPM stage.

## D  FURTHER STUDY ON COMPONENT I

We conduct a series of supplementary analyses to unpack this question. In Section D.1, we test the stability of extracted rules by varying the number of top-$k$ trees and find that rule quality remains robust across settings. In Section D.2, we compare rule formats and show that explicit *if–then* clauses provide clearer guidance than natural-language paraphrases. In Section D.3, we examine rule denoising strategies and demonstrate that self-consistency aggregation yields more reliable rules than single-pass or CoT prompting. In Section D.4, we validate transferability across different LLM backbones, confirming that rule guidance consistently improves synthetic data quality regardless of model scale. In Section D.5, a case study illustrates how noisy tree paths are distilled into compact and interpretable rules through merging and denoising, highlighting both robustness and interpretability. Together, these studies confirm that rules enhance data quality by providing stable, precise, and broadly applicable generation guidance.

Table 6: Rule Compliance Rate (RCR) Results for different $k$ values ($D_{\text{test}}$ and $D_{\text{train}}$).

|  | $k=1$ | | $k=2$ | | $k=3$ | | $k=5$ | | $k=10$ | |
|---|---|---|---|---|---|---|---|---|---|---|
|  | $D_{\text{test}}$ | $D_{\text{train}}$ | $D_{\text{test}}$ | $D_{\text{train}}$ | $D_{\text{test}}$ | $D_{\text{train}}$ | $D_{\text{test}}$ | $D_{\text{train}}$ | $D_{\text{test}}$ | $D_{\text{train}}$ |
| Disease (N=30) | 63.44 | 83.33 | 36.38 | 53.33 | 29.79 | 40.00 | 44.61 | 50.00 | 72.99 | 63.33 |
| GPA (N=30) | 62.45 | 70.00 | 62.45 | 70.00 | 58.17 | 66.67 | 62.57 | 70.00 | 63.76 | 60.00 |
| Student (N=30) | 46.12 | 56.67 | 28.83 | 50.00 | 26.53 | 50.00 | 50.66 | 73.33 | 36.45 | 30.00 |
| Disease (N=60) | 60.67 | 76.67 | 34.92 | 35.00 | 43.04 | 53.33 | 26.41 | 28.33 | 50.93 | 56.67 |
| GPA (N=60) | 41.90 | 36.67 | 42.02 | 56.67 | 58.02 | 70.00 | 49.01 | 65.00 | 49.70 | 58.33 |
| Student (N=60) | 17.72 | 13.33 | 54.77 | 48.33 | 34.46 | 36.67 | 43.07 | 35.00 | 57.95 | 61.67 |
| Disease (N=90) | 50.56 | 70.00 | 34.61 | 46.67 | 69.18 | 80.00 | 13.67 | 17.78 | 61.10 | 73.33 |
| GPA (N=90) | 50.61 | 57.78 | 50.74 | 57.78 | 74.85 | 70.00 | 66.12 | 62.22 | 54.00 | 48.89 |
| Student (N=90) | 31.58 | 26.67 | 26.67 | 22.22 | 31.13 | 27.78 | 55.65 | 50.00 | 49.38 | 51.11 |
| Disease (N=120) | 51.21 | 63.33 | 27.16 | 41.67 | 30.26 | 39.17 | 28.63 | 38.33 | 28.78 | 38.33 |
| GPA (N=120) | 40.98 | 44.17 | 48.50 | 48.33 | 60.74 | 41.67 | 54.52 | 39.11 | 64.73 | 54.62 |
| Student (N=120) | 29.57 | 31.67 | 65.82 | 63.33 | 42.58 | 43.33 | 32.36 | 34.17 | 28.97 | 34.17 |

### D.1  EFFECT OF TOP-k TREES IN RULE EXTRACTION.

To examine the robustness of the extracted rules, we vary the number of top-$k$ trees used for rule extraction ($k=1, 2, 3, 5, 10$). Across all settings, the resulting rule-compliance rates (RCR) remain stable, suggesting that the induced rules capture consistent feature–label dependencies that are not sensitive to the choice of $k$. While $k=1$ or $2$ already achieve similar RCR, we adopt $k=3$ as a default since it provides broader rule coverage than very small $k$ while avoiding the additional overhead of larger $k$ values. This balance ensures that the extracted rules are both robust and efficient for downstream prompting.

### D.2  DIFFERENT RULE FORMAT COMPARISON.

To gauge whether the *explicit* "if–then" representation is essential to the success of Rule-Guided Generation, we recast every Random-Forest path into two formats: (i) its original "if–then" clause and (ii) a concise natural-language paraphrase, which automatically produced by prompting the LLM to restate each path in natural language. These two formats reflect two distinct rule-extraction schemes. A side-by-side example of the two rule forms is shown in Figure 7. As shown in Table 7, *if–then* rules outperform both natural-language rules and the No-Rule baseline, demonstrating the benefit of symbolic structure. While both rule-based approaches surpass the baseline, the symbolic form delivers more reliable performance. This advantage stems from two key factors: (i) Random Forests extract concise and faithful feature–label dependencies even in low-data settings, and (ii) the *if–then* format retains explicit numeric boundaries and dependent constraints, unlike natural language, which tends to weaken precision (Xu et al., 2024b). By guiding generation through explicit

| Natraul Language Rule Form | "If-Then" Rule Form |
|---|---|
| - There is a noticeable trend where patients diagnosed with Alzheimer's often have lower MMSE scores. This trend is particularly apparent among older patients, suggesting a strong association between advanced age, decreased cognitive function, and an Alzheimer's diagnosis. The lower functional assessment scores in these individuals further support this link, indicating that cognitive decline is a key factor in distinguishing those with Alzheimer's. | - If Diagnosis = 1:
 - Then MMSE ≤ 16.85 and FunctionalAssessment < 6.90
 - If Diagnosis = 0:
 - Then MMSE > 18.85 and FunctionalAssessment > 6.90 |

Figure 7: Illustrative "if–then" Form and its Natural-Language paraphrase derived from the *Disease* dataset ($N$=30).

Table 7: Performance under different rule representations ($n = 30$).

Table 8: Performance under different *rule denoising* strategies ($n = 30$).

|  | No Rule | Natural Language | "if-then" Form |
|---|---|---|---|
| Disease | $49.13 \pm 1.2$ | $57.54 \pm 1.7$ | $\mathbf{59.61 \pm 1.5}$ |
| GPA | $24.80 \pm 4.0$ | $37.78 \pm .60$ | $\mathbf{41.21 \pm .97}$ |
| Student | $-0.11 \pm .12$ | $-0.79 \pm .53$ | $\mathbf{0.37 \pm .02}$ |

|  | Single-Pass | COT | Self-Consistency |
|---|---|---|---|
| Disease | $54.26 \pm 3.2$ | $58.46 \pm .98$ | $\mathbf{59.61 \pm 1.5}$ |
| GPA | $24.80 \pm 4.0$ | $38.48 \pm .82$ | $\mathbf{41.21 \pm .97}$ |
| Student | $0.35 \pm .06$ | $0.35 \pm .02$ | $\mathbf{0.37 \pm .02}$ |

symbolic rules, the LLM is directed toward semantically coherent subspaces, resulting in higher-quality samples.

### D.3 ROBUSTNESS OF RULE DENOISING STRATEGY.

We evaluate how different Rule Denoising strategies affect data quality. We compare our proposed *Self-Consistency Rule Denoising* (described in Section 3.2) against two alternatives: (i) **Single-Pass**, which denoise rules in one step without verification, and (ii) **Chain-of-Thought (CoT)** prompting, which guides the LLM to reason step-by-step during rule denoising (Wei et al., 2022). Results in Table 8 reveal that both **Single-Pass** and **CoT** aggregation yield less consistent performance across datasets. Their reliance on one-shot reasoning makes them vulnerable to local inconsistencies and stochastic behavior in LLM outputs. In contrast, self-consistency aggregation enforces cross-run agreement and filters out unstable logic fragments, leading to more robust rule sets and better downstream fidelity.

### D.4 RULE-BASED METHODS GENERALIZE ACROSS LLMS

To evaluate the robustness of *Rule-Guided Generation* in different LLM backbones, we compare performance using the MLE under varying combinations of rules generators and data generators. The result as shown in Table 9, across all tested LLMs which ranging from mid-sized (Qwen2.5-14b) to larger models (Qwen2.5-32b and GPT-4o), the introduction of association rules consistently improves the quality of synthetic data. Notably, even when rules are extracted by weaker models (e.g., Qwen2.5-32b), stronger models like GPT-4o still benefit from them—highlighting that rule-

Table 9: Main results on benchmark datasets ($n$=30). We report **F1 score** for classification tasks and **$R^2$** for regression tasks. The best result in each row is shown in **bold**.

| Original Data | | Rules Generator | Data Generator | | |
|---|---|---|---|---|---|
| Datasets | Real Data | | Qwen2.5-14b-Instruct | Qwen2.5-32b-Instruct | GPT-4o-0806 |
| Disease | $93.68 \pm 1.4$ | No Rules | $59.52 \pm 1.9$ | $65.12 \pm 2.3$ | $62.26 \pm 3.2$ |
| | | Qwen2.5-32b-Instruct | $66.34 \pm 1.4$ | $70.16 \pm .80$ | $67.38 \pm .85$ |
| | | GPT-4o-0806 | $\mathbf{70.21 \pm 1.9}$ | $67.74 \pm 1.4$ | $64.04 \pm 2.7$ |
| GPA | $47.29 \pm .90$ | No Rules | $20.67 \pm .87$ | $30.23 \pm 1.4$ | $27.58 \pm 2.3$ |
| | | Qwen2.5-32b-Instruct | $40.62 \pm 1.4$ | $39.13 \pm 3.4$ | $\mathbf{47.07 \pm 0.5}$ |
| | | GPT-4o-0806 | $42.25 \pm 1.4$ | $31.99 \pm 2.9$ | $44.34 \pm 0.6$ |
| Student | $0.67 \pm .07$ | No Rules | $0.25 \pm .04$ | $0.32 \pm .03$ | $-1.29 \pm .27$ |
| | | Qwen2.5-32b-Instruct | $0.30 \pm .03$ | $\mathbf{0.33 \pm .03}$ | $0.25 \pm .01$ |
| | | GPT-4o-0806 | $0.18 \pm .05$ | $-0.38 \pm .16$ | $-0.01 \pm .14$ |

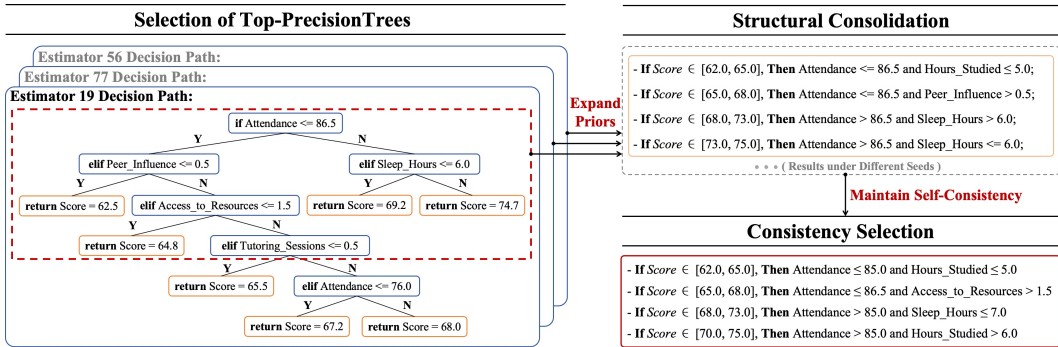

Figure 8: Case study for Component I on the *Student* dataset. Noisy tree paths are denoised into a self-consistent symbolic formulas set and later guides data generation.

based guidance contributes independently of LLMs capacity. This demonstrates the generality and transferability of our design.

### D.5 ILLUSTRATIVE EXAMPLE OF RULE GENERALIZATION & DENOISING.

To illustrate the practical functioning of **Component I**, we present a case study demonstrating how symbolic rules are distilled into interpretable *if–then* forms through rule merging and aggregation. Intermediate outcomes are visualized in Figure 8. In this example, decision trees from the trained random forest exhibit conflicting paths—for example, assigning different $Exam\_Score$ values to overlapping input regions such as $Attendance \leq 86.5$ and $Attendance \leq 76.0$. The merging phase addresses these inconsistencies by consolidating noisy rule fragments into coherent patterns, such as those involving $Sleep\_Hours$. To improve robustness, the training and generalization process is repeated under multiple random seeds. The denoising step then retains only those patterns that appear consistently across runs, thereby filtering out unstable conditions and enforcing self-consistency. This case highlights two key strengths of Component I: (1) it distills noisy and fragmented tree logic into compact rules that capture meaningful relationships in low-data regimes; (2) it produces one concise association rule per segment, enhancing both interpretability and generation quality.

## E  Further Study on Component II

To further examine *how filtering balances the distribution*, we study the behavior of **Component II** under different control settings. Specifically, we analyze (i) the effect of the log-scaled retention function on pruning schedules, and (ii) the stability of the Gini coefficient under varying generation sizes. These studies show that Gini-based filtering prunes aggressively only when redundancy is severe while maintaining diversity in balanced regimes, and that Gini values quickly stabilize as generation size grows. Together, these results confirm that our filtering strategy provides a robust and scale-invariant mechanism for mitigating localized redundancy and restoring distributional balance. In following experiments, we fix the backbone settings (GPT-4o-0806 as Rule Generator and GPT-3.5-turbo-1106 as Data Generator) and evaluate results via MLE score. We report **F1 score** for *Disease* and *GPA*, and **$R^2$** for *Student*.

### E.1  Impact on the Log-Scaled Retention Function.

Using the log-scaled mapping in (3), we vary $G(p)$ across its empirical range and record the resulting $\text{ratio}_{prune}$ as well as downstream F1. Figure 9 (Left) shows that performance peaks at intermediate retention, validating that the Gini-driven schedule prunes aggressively only when redundancy is severe, while preserving diversity in more balanced settings. We observe that the Gini values across datasets tend to lie in a relatively narrow and stable range, which contributes to the robustness of the fitted retention schedule. As a result, the coefficients $A = 0.15$ and $B = 0.55$, derived from cross-dataset regression, generalize well without tuning.

### E.2  Stability Under Varying Generation Sizes

We next test whether Gini is sensitive to the number of generated samples, since an unstable control signal would undermine filtering. Figure 9 (Right) shows that the Gini coefficient stabilizes after roughly 1,000 samples across tasks, with only minor fluctuations thereafter. This indicates that dominant distributional modes emerge early in generation, and that Gini provides a consistent, scale-invariant signal for downstream filtering decision.

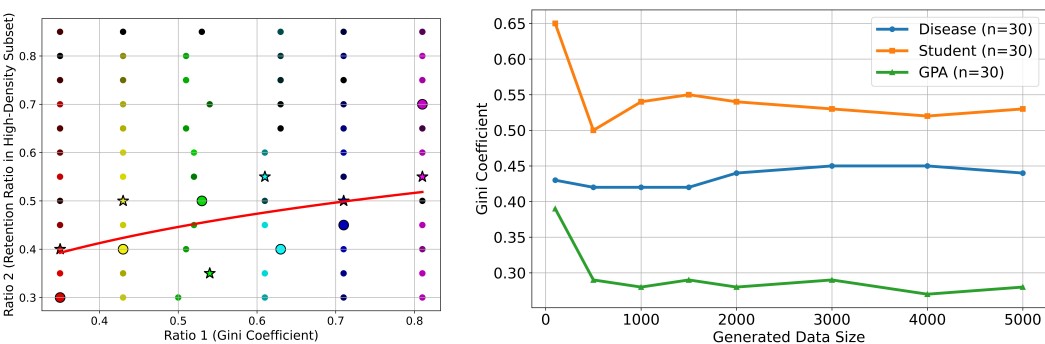

Figure 9: (Left) Scatter plot of $G(p)$ (Gini coefficient) versus $Ratio_{prune}$. Point color indicates downstream model performance after filtering, with lighter colors representing higher performance and darker colors indicating lower performance. In each group, the star ($\star$) marks the best-performing point and the circle ($\circ$) marks the second-best. (Right) Gini coefficient under different synthetic data sizes.

# F EXTENDED EXPERIMENT

Table 10: Main results in Non-LLM methods. We report $F_1$ **score** for classification tasks and $\mathbf{R^2}$ for regression tasks. All synthetic data are evaluated using the same **XGBoost** downstream model. All values are reported as **mean ± standard deviation** (computed over multiple runs). Unseen datasets are highlighted in blue, and seen datasets are highlighted in red.

| | Original Data | Non-LLM Methods | | | LLM-Based Methods | |
|---|---|---|---|---|---|---|
| **Datasets** | **Real** | CTGAN | DRL | TABSYN | GREAT | EPIC |
| **N = 30** | | | | | | |
| Disease (N=30) | 93.68±1.4 | 44.58±4.7 | 39.49±0.31 | 54.79±2.7 | 46.54±1.7 | 32.01±8.2 |
| Game (N=30) | 86.0±0.73 | 32.03±4.9 | 33.54±6.33 | 44.13±1.4 | 53.65±3.2 | 13.93±1.2 |
| Apple (N=30) | 86.60±1.1 | 47.21±1.8 | 51.33±0.46 | 32.32±1.3 | 58.91±2.3 | 56.10±0.7 |
| GPA (N=30) | 47.29±0.90 | 15.76±3.5 | 14.46±1.54 | 19.22±2.5 | 31.57±1.9 | 32.03±2.8 |
| Student (N=30) | 0.67±0.07 | -0.91±1.12 | -0.01±0.01 | 0.14±0.04 | 0.21±0.03 | 0.35±0.05 |
| Farm (N=30) | -0.04±0.03 | -0.51±0.06 | -0.07±0.04 | -0.38±0.05 | -1.03±2.1 | -0.68±0.06 |
| Adult (N=30) | 76.92±0.68 | 50.47±4.1 | 46.59±2.44 | 62.83±3.2 | 67.45±1.3 | 61.94±3.3 |
| Heart (N=30) | 86.71±1.7 | 48.16±8.8 | 38.66±2.29 | 79.40±1.3 | 80.74±1.3 | 81.20±1.3 |
| **Avg Rank** | - | 7.9 | 7.1 | 6.1 | 4.8 | 6 |
| **N = 60** | | | | | | |
| Disease (N=60) | 93.68±1.4 | 30.03±3.2 | 38.25±0.22 | 65.34±9.3 | 52.80±3.1 | 44.05±7.8 |
| Game (N=60) | 86.0±0.73 | 31.83±2.4 | 30.10±1.07 | 61.10±1.1 | 54.76±2.6 | 13.16±0 |
| Apple (N=60) | 86.60±1.1 | 40.40±1.3 | 49.23±0.13 | 27.07±0.68 | 60.33±1.6 | 70.87±0.85 |
| GPA (N=60) | 47.29±0.90 | 18.11±3.3 | 15.64±1.88 | 28.48±1.8 | 35.60±2.4 | 32.19±3.5 |
| Student (N=60) | 0.67±0.07 | -0.07±0.05 | -0.04±0.08 | -0.14±0.14 | 0.02±0.09 | -0.63±0.19 |
| Farm (N=60) | -0.04±0.03 | -0.22±0.05 | -0.16±0.05 | -0.17±0.04 | -0.16±0.04 | -0.21±0.03 |
| Adult (N=60) | 76.92±0.68 | 47.55±5.1 | 49.02±1.34 | 63.58±1.6 | 69.70±0.78 | 62.78±1.5 |
| Heart (N=60) | 86.71±1.7 | 49.77±9.3 | 38.14±2.85 | 81.35±1.2 | 80.64±0.82 | 77.55±1.6 |
| **Avg Rank** | - | 7.8 | 7 | 5.3 | 4.3 | 6.6 |
| **N = 90** | | | | | | |
| Disease (N=90) | 93.68±1.4 | 47.18±3.5 | 39.04±0.00 | 69.90±2.0 | 65.32±1.4 | 30.03±3.2 |
| Game (N=90) | 86.0±0.73 | 33.67±2.5 | 27.81±2.93 | 65.93±1.2 | 61.17±3.0 | 13.16±0 |
| Apple (N=90) | 86.60±1.1 | 43.08±1.2 | 38.67±1.59 | 20.69±0.61 | 64.23±2.6 | 73.67±1.0 |
| GPA (N=90) | 47.29±0.90 | 14.91±2.3 | 12.24±0.90 | 34.51±1.3 | 36.19±1.1 | 16.28±1.98 |
| Student (N=90) | 0.67±0.07 | -0.02±0.01 | -0.03±0.05 | -0.12±0.10 | 0.22±0.12 | 0.32±0.07 |
| Farm (N=90) | -0.04±0.03 | -0.34±0.03 | -0.05±0.04 | -0.15±0.03 | -0.98±0.71 | -0.16±0.04 |
| Adult (N=90) | 76.92±0.68 | 41.67±2.4 | 47.47±2.36 | 69.77±0.68 | 71.52±0.84 | 66.73±1.3 |
| Heart (N=90) | 86.71±1.7 | 42.04±4.1 | 39.03±2.85 | 81.43±1.3 | 81.05±0.88 | 77.65±1.7 |
| **Avg Rank** | - | 7.4 | 7.1 | 4.6 | 5.1 | 6.3 |
| **N = 120** | | | | | | |
| Disease (N=120) | 93.68±1.4 | 45.30±3.51 | 73.92±3.30 | 62.37±1.25 | 55.70±3.95 | 55.37±10.36 |
| Game (N=120) | 86.0±0.73 | 26.95±1.52 | 31.04±2.48 | 63.50±0.78 | 66.32±5.11 | 48.99±2.41 |
| Apple (N=120) | 86.60±1.1 | 39.83±3.39 | 34.29±0.52 | 80.54±0.49 | 60.91±2.12 | 79.16±0.64 |
| GPA (N=120) | 47.29±0.90 | 16.58±1.41 | 10.63±1.75 | 38.57±2.30 | 51.29±1.6 | 46.61±3.7 |
| Student (N=120) | 0.67±0.07 | 0.00±0.01 | 0.01±0.03 | 0.41±0.03 | 0.23±0.04 | 0.29±0.07 |
| Farm (N=120) | -0.04±0.03 | -0.05±0.04 | -0.01±0.01 | -0.22±0.03 | -0.18±0.04 | -0.30±0.03 |
| Adult (N=120) | 76.92±0.68 | 48.12±5.58 | 39.22±6.96 | 68.87±0.41 | 72.58±1.22 | 68.35±2.10 |
| Heart (N=120) | 86.71±1.7 | 46.29±3.1 | 38.29±0.31 | 83.80±0.51 | 84.17±0.91 | 71.33±6.33 |
| **Avg Rank** | - | 7.6 | 7.1 | 3.9 | 3.9 | 5.6 |

Table 11: Main results in LLM-based methods. We report $F_1$ **score** for classification tasks and $\mathbf{R^2}$ for regression tasks. All synthetic data are evaluated using the same **XGBoost** downstream model. All values are reported as **mean ± standard deviation** (computed over multiple runs). Unseen datasets are highlighted in blue, and seen datasets are highlighted in red.

| Original Data | | LLM-Based Methods | | | |
|---|---|---|---|---|---|
| **Datasets** | **Real** | **CLLM** | **I \ II** | **II \ I** | **I+II** |
| N = 30 | | | | | |
| Disease (N=30) | 93.68±1.4 | 61.89±2.1 | 59.61±1.5 | 64.44±1.7 | 70.22±0.83 |
| Game (N=30) | 86.0±0.73 | 54.12±2.7 | 56.44±2.2 | 45.00±1.3 | 59.13±2.8 |
| Apple (N=30) | 86.60±1.1 | 58.18±1.5 | 57.80±1.3 | 58.66±0.12 | 59.43±1.0 |
| GPA (N=30) | 47.29±0.90 | 40.17±0.46 | 41.21±0.97 | 35.88±1.7 | 43.14±0.59 |
| Student (N=30) | 0.67±0.07 | -0.11±0.12 | 0.37±0.02 | 0.08±0.06 | 0.38±0.02 |
| Farm (N=30) | -0.04±0.03 | -0.29±0.04 | -0.44±0.06 | -0.23±0.06 | -0.30±0.04 |
| Adult (N=30) | 76.92±0.68 | 73.11±0.65 | 73.04±0.59 | 73.15±0.18 | 73.91±0.37 |
| Heart (N=30) | 86.71±1.7 | 80.47±0.84 | 75.17±4.7 | 80.00±0.24 | 80.14±0.33 |
| **Avg Rank** | - | 3.8 | 4 | 3.6 | **1.8** |
| N = 60 | | | | | |
| Disease (N=60) | 93.68±1.4 | 66.86±1.1 | 75.65±2.5 | 62.58±2.2 | 72.41±0.77 |
| Game (N=60) | 86.0±0.73 | 67.54±1.0 | 61.39±2.6 | 59.81±1.1 | 70.87±0.42 |
| Apple (N=60) | 86.60±1.1 | 68.60±1.0 | 58.71±1.7 | 69.92±1.1 | 66.70±1.5 |
| GPA (N=60) | 47.29±0.90 | 33.17±1.1 | 44.34±0.91 | 21.57±1.8 | 44.58±0.56 |
| Student (N=60) | 0.67±0.07 | -0.48±0.19 | 0.27±0.03 | -0.15±0.04 | 0.34±0.07 |
| Farm (N=60) | -0.04±0.03 | -0.30±0.04 | -0.21±0.03 | -0.12±0.02 | -0.15±0.02 |
| Adult (N=60) | 76.92±0.68 | 73.94±0.54 | 73.28±0.85 | 72.87±0.50 | 71.48±0.60 |
| Heart (N=60) | 86.71±1.7 | 80.33±0.85 | 81.37±0.96 | 78.15±0.53 | 80.36±0.03 |
| **Avg Rank** | - | 4.3 | 2.9 | 4.4 | **2.4** |
| N = 90 | | | | | |
| Disease (N=90) | 93.68±1.4 | 74.04±1.6 | 69.74±1.5 | 65.47±1.2 | 76.45±1.1 |
| Game (N=90) | 86.0±0.73 | 59.97±2.3 | 59.44±2.7 | 41.77±1.6 | 62.17±2.7 |
| Apple (N=90) | 86.60±1.1 | 73.43±0.90 | 65.23±2.6 | 72.78±1.1 | 74.88±1.5 |
| GPA (N=90) | 47.29±0.90 | 41.79±1.1 | 47.21±1.3 | 31.56±0.87 | 48.89±0.56 |
| Student (N=90) | 0.67±0.07 | -0.69±0.13 | 0.34±0.09 | -0.27±0.09 | 0.47±0.09 |
| Farm (N=90) | -0.04±0.03 | -0.16±0.02 | -0.15±0.03 | -0.14±0.02 | -0.13±0.02 |
| Adult (N=90) | 76.92±0.68 | 74.11±1.36 | 78.45±0.97 | 73.75±0.51 | 74.10±0.62 |
| Heart (N=90) | 86.71±1.7 | 81.23±1.0 | 82.44±0.86 | 78.74±1.4 | 80.00±0.03 |
| **Avg Rank** | - | 4 | 3 | 5.3 | **2** |
| N = 120 | | | | | |
| Disease (N=120) | 93.68±1.4 | 78.30±1.44 | 78.97±3.13 | 61.82±2.57 | 81.96±0.64 |
| Game (N=120) | 86.0±0.73 | 61.32±1.16 | 45.48±1.62 | 54.76±2.01 | 61.67±1.40 |
| Apple (N=120) | 86.60±1.1 | 70.66±0.85 | 70.62±3.12 | 68.53±0.50 | 74.96±1.00 |
| GPA (N=120) | 47.29±0.90 | 45.20±2.1 | 40.42±1.02 | 46.95±0.5 | 47.77±0.81 |
| Student (N=120) | 0.67±0.07 | 0.22±0.03 | 0.21±0.04 | 0.17±0.06 | 0.29±0.04 |
| Farm (N=120) | -0.04±0.03 | -0.23±0.03 | -0.10±0.03 | -0.29±0.03 | -0.07±0.02 |
| Adult (N=120) | 76.92±0.68 | 71.73±0.82 | 73.36±0.67 | 73.19±0.65 | 72.93±0.74 |
| Heart (N=120) | 86.71±1.7 | 75.24±0.72 | 84.97±1.08 | 75.02±0.97 | 75.93±0.84 |
| **Avg Rank** | - | 4.8 | 4 | 5.4 | **2.6** |

Table 12: Comparison between Best Non-LLM baselines, LLM-based generators, and TABPFN-GEN. We report $F_1$ **score** for classification tasks and $\mathbf{R^2}$ for regression tasks. Unseen datasets are highlighted in blue. All synthetic data are evaluated using the same **XGBoost** downstream model. All values are reported as **mean ± standard deviation** (computed over multiple runs). The best result in each row is shown in **bold**, and the second-best result is underlined.

| Datasets | Real | TABSYN | EPIC | CLLM | I \ II | II \ I | I+II | TABPFNGEN |
|---|---|---|---|---|---|---|---|---|
| **N = 30** | | | | | | | | |
| Disease | 93.68±1.4 | 54.79±2.7 | 32.01±8.2 | 61.89±2.1 | 59.61±1.5 | 64.44±1.7 | **70.22±0.83** | 61.38±0.45 |
| Game | 86.0±0.73 | 44.13±1.4 | 13.93±1.2 | 54.12±2.7 | 56.44±2.2 | 45.00±1.3 | 59.13±2.8 | **70.43±1.81** |
| GPA | 47.29±0.90 | 19.22±2.5 | 32.03±2.8 | 40.17±0.46 | 41.21±0.97 | 35.88±1.7 | **43.14±0.59** | 41.40±0.92 |
| Student | 0.67±0.07 | 0.14±0.04 | 0.35±0.05 | -0.11±0.12 | 0.37±0.02 | 0.08±0.06 | **0.38±0.02** | 0.37±0.01 |
| **Avg Rank** | - | 6 | 6 | 4.3 | 3.3 | 4.8 | **1.3** | 2.3 |
| **N = 60** | | | | | | | | |
| Disease | 93.68±1.4 | 65.34±9.3 | 44.05±7.8 | 66.86±1.1 | **75.65±2.5** | 62.58±2.2 | 72.41±0.77 | 71.88±0.62 |
| Game | 86.0±0.73 | 61.10±1.1 | 13.16±0 | 67.54±1.0 | 61.39±2.6 | 59.81±1.1 | 70.87±0.42 | **78.28±0.99** |
| GPA | 47.29±0.90 | 28.48±1.8 | 32.19±3.5 | 33.17±1.1 | 44.34±0.91 | 21.57±1.8 | **44.58±0.56** | 42.81±0.61 |
| Student | 0.67±0.07 | -0.14±0.14 | -0.63±0.19 | -0.48±0.19 | 0.27±0.03 | -0.15±0.04 | 0.34±0.07 | **0.46±0.01** |
| **Avg Rank** | - | 5.5 | 5.5 | 3.3 | 3 | 6 | **2.3** | 2.5 |
| **N = 90** | | | | | | | | |
| Disease | 93.68±1.4 | 69.90±2.0 | 30.03±3.2 | 74.04±1.6 | 69.74±1.5 | 65.47±1.2 | **76.45±1.1** | 75.55±0.36 |
| Game | 86.0±0.73 | **65.93±1.2** | 13.16±0 | 59.97±2.3 | 59.44±2.7 | 41.77±1.6 | 62.17±2.7 | 60.12±0.95 |
| GPA | 47.29±0.90 | 34.51±1.3 | 16.28±1.98 | 41.79±1.1 | 47.21±1.3 | 31.56±0.87 | 48.89±0.56 | **49.77±0.66** |
| Student | 0.67±0.07 | -0.12±0.10 | 0.32±0.07 | 0.11±0.13 | 0.34±0.09 | 0.27±0.09 | **0.47±0.09** | 0.44±0.05 |
| **Avg Rank** | - | 4 | 6.3 | 4.5 | 4 | 5.8 | **1.5** | 2 |
| **N = 120** | | | | | | | | |
| Disease | 93.68±1.4 | 62.37±1.25 | 55.37±10.36 | 78.30±1.44 | 78.97±3.13 | 61.82±2.57 | **81.96±0.64** | 78.85±0.40 |
| Game | 86.0±0.73 | 63.50±0.78 | 48.99±2.41 | 61.32±1.16 | 45.48±1.62 | 54.76±2.01 | 61.67±1.40 | **73.93±0.41** |
| GPA | 47.29±0.90 | 38.57±2.30 | 46.61±3.7 | 45.20±2.1 | 40.42±1.02 | 46.95±0.5 | **47.77±0.81** | 44.78±0.71 |
| Student | 0.67±0.07 | 0.41±0.03 | 0.29±0.07 | 0.22±0.03 | 0.21±0.04 | 0.17±0.06 | 0.29±0.04 | **0.47±0.03** |
| **Avg Rank** | - | 4 | 4.8 | 4.3 | 5.3 | 5 | **2** | 2.5 |
| **N = 160** | | | | | | | | |
| Disease | 93.68±1.4 | 72.25±1.33 | 68.75±2.71 | 69.86±2.51 | 58.60±2.09 | 63.60±1.73 | 71.66±1.75 | **76.45±0.24** |
| Game | 86.0±0.73 | 69.91±0.64 | 49.69±4.99 | 48.18±3.40 | 39.78±1.33 | 54.45±1.50 | 62.79±1.13 | **74.08±0.35** |
| GPA | 47.29±0.90 | 39.37±0.81 | 29.08±1.04 | 31.07±1.83 | 44.49±1.76 | 41.01±2.41 | 44.65±0.63 | **45.62±0.75** |
| Student | 0.67±0.07 | 0.32±0.07 | 0.43±0.10 | 0.22±0.07 | 0.30±0.17 | 0.37±0.12 | **0.46±0.13** | 0.43±0.02 |
| **Avg Rank** | - | 3.5 | 4.8 | 5.8 | 5.8 | 4.5 | 2.3 | **1.3** |
| **N = 200** | | | | | | | | |
| Disease | 93.68±1.4 | 64.53±1.49 | 55.30±6.14 | 73.72±1.98 | 68.58±2.13 | 71.96±2.86 | 74.25±1.38 | **78.07±0.84** |
| Game | 86.0±0.73 | **77.92±0.69** | 55.19±6.76 | 51.19±2.06 | 44.68±3.78 | 61.56±0.68 | 61.98±1.52 | 69.18±2.03 |
| GPA | 47.29±0.90 | 41.59±0.86 | 40.88±0.94 | 36.40±3.18 | 43.92±1.42 | 38.73±0.00 | 44.68±0.51 | **46.26±0.90** |
| Student | 0.67±0.07 | 0.39±0.11 | 0.51±0.04 | 0.24±0.06 | 0.50±0.08 | 0.20±0.07 | **0.52±0.02** | 0.50±0.01 |
| **Avg Rank** | - | 4 | 4.8 | 5.5 | 4.5 | 5.3 | 2 | **1.8** |
| **N = 300** | | | | | | | | |
| Disease | 93.68±1.4 | 77.26±1.18 | 35.37±2.47 | 67.31±1.59 | 68.65±4.11 | 66.02±1.73 | 73.31±2.82 | **78.96±0.51** |
| Game | 86.0±0.73 | **77.64±0.56** | 54.23±4.23 | 46.69±2.05 | 47.58±3.79 | 47.25±2.47 | 61.67±1.23 | 71.90±1.61 |
| GPA | 47.29±0.90 | 39.86±1.29 | 37.77±0.63 | 39.67±2.14 | 42.98±0.05 | 43.98±0.96 | **46.72±1.12** | 46.09±1.44 |
| Student | 0.67±0.07 | 0.54±0.01 | 0.44±0.03 | -0.03±0.06 | 0.50±0.05 | -0.17±0.06 | **0.55±0.01** | 0.54±0.01 |
| **Avg Rank** | - | 2.5 | 5.8 | 6.3 | 4.3 | 5.3 | 2 | **1.8** |
| **N = 500** | | | | | | | | |
| Disease | 93.68±1.4 | 69.93±1.41 | 65.26±3.15 | 67.56±2.83 | 61.41±3.90 | 68.55±2.44 | 72.48±1.97 | **78.52±0.26** |
| Game | 86.0±0.73 | **80.33±1.26** | 61.38±5.18 | 53.88±1.77 | 69.37±2.33 | 65.33±1.19 | 75.95±1.57 | 79.60±0.89 |
| GPA | 47.29±0.90 | 40.17±1.20 | 36.35±0.95 | 30.14±2.30 | 37.90±1.94 | 35.93±0.98 | 40.30±0.65 | **43.13±0.74** |
| Student | 0.67±0.07 | **0.57±0.01** | 0.54±0.02 | -0.28±0.08 | 0.46±0.04 | 0.03±0.22 | 0.52±0.01 | 0.56±0.01 |
| **Avg Rank** | - | 2 | 5 | 6.3 | 5 | 5.5 | 2.8 | **1.5** |

Table 13: Comparison between Non-LLM baselines, LLM-based generators, and TABPFNGEN. All results are evaluated with the **TABPFN** downstream model and reported as mean±standard deviation. Unseen datasets are highlighted in blue. The best result in each row is shown in **bold**, and the second-best result is underlined.

| Datasets | Real | TABSYN | EPIC | CLLM | I(w/oII) | II(w/oI) | I+II | TABPFNGEN |
|---|---|---|---|---|---|---|---|---|
| **N = 30** | | | | | | | | |
| Disease | 96.12±0.28 | 55.35±2.26 | 33.69±9.38 | 68.09±1.89 | 55.50±0.40 | 61.34±1.78 | **69.45±0.23** | 56.75±0.83 |
| Game | 90.11±0.31 | 38.31±2.11 | 37.33±5.61 | 58.54±1.72 | 50.80±2.35 | 51.66±0.00 | 51.36±0.81 | **69.18±2.03** |
| GPA | 71.07±1.48 | 16.37±1.84 | 30.38±2.28 | 40.26±0.50 | 41.84±1.27 | 36.55±0.00 | **42.33±0.64** | 39.12±0.39 |
| Student | 0.73±0.00 | 0.18±0.05 | 0.28±0.04 | -0.26±0.14 | 0.40±0.02 | 0.15±0.00 | 0.44±0.01 | **0.54±0.01** |
| **Avg Rank** | - | 6 | 6 | 3.5 | 3.8 | 4.3 | **2** | 2.5 |
| **N = 60** | | | | | | | | |
| Disease | 96.12±0.28 | 62.56±0.94 | 27.33±1.53 | **70.84±2.19** | 62.66±1.61 | 63.70±0.65 | 68.50±1.00 | 69.22±1.27 |
| Game | 90.11±0.31 | 57.03±1.76 | 50.84±4.16 | 65.79±1.49 | 59.88±2.70 | 54.43±0.95 | 63.25±0.00 | **75.87±1.18** |
| GPA | 71.07±1.48 | 29.09±0.85 | 38.26±1.50 | 36.20±2.81 | 45.12±0.64 | 27.30±0.00 | **48.89±0.25** | 47.45±0.87 |
| Student | 0.73±0.00 | 0.11±0.12 | 0.29±0.05 | -0.48±0.11 | 0.27±0.05 | -0.03±0.04 | 0.35±0.02 | **0.54±0.01** |
| **Avg Rank** | - | 5.5 | 5.3 | 3.8 | 4 | 5.8 | 2.3 | **1.5** |
| **N = 90** | | | | | | | | |
| Disease | 96.12±0.28 | 71.90±1.24 | 28.39±2.57 | **77.01±1.10** | 54.44±2.97 | 67.25±2.29 | 62.84±1.06 | 73.04±2.0 |
| Game | 90.11±0.31 | 39.48±1.05 | 54.31±2.00 | 66.10±2.21 | 48.52±1.77 | 48.52±1.77 | 67.46±2.31 | **72.83±0.63** |
| GPA | 71.07±1.48 | 35.49±1.54 | 14.58±1.92 | 46.62±0.79 | 48.48±1.55 | 37.89±1.93 | 49.49±0.55 | 52.33±0.43 |
| Student | 0.73±0.00 | 0.21±0.09 | 0.38±0.05 | -0.27±0.15 | 0.48±0.03 | 0.14±0.04 | **0.53±0.01** | 0.53±0.01 |
| **Avg Rank** | - | 5.3 | 5.8 | 3.8 | 4 | 5.3 | 2.5 | **1.3** |
| **N = 120** | | | | | | | | |
| Disease | 96.12±0.28 | 64.04±1.55 | 31.89±9.27 | **71.33±1.31** | 62.33±4.82 | 63.08±2.65 | 64.65±1.95 | 68.87±2.74 |
| Game | 90.11±0.31 | 57.47±0.96 | 53.76±1.43 | 74.16±0.78 | 60.56±2.40 | 62.54±0.00 | 69.62±0.70 | **74.81±0.57** |
| GPA | 71.07±1.48 | 37.13±4.26 | 44.32±1.83 | 47.43±2.11 | 42.36±1.23 | 38.67±2.03 | 41.06±1.76 | **57.26±1.10** |
| Student | 0.73±0.00 | 0.05±0.08 | 0.54±0.02 | -0.50±0.16 | 0.46±0.07 | 0.04±0.03 | **0.56±0.02** | 0.53±0.01 |
| **Avg Rank** | - | 5.5 | 4.8 | 3 | 4.8 | 5.3 | 3 | **1.8** |
| **N = 160** | | | | | | | | |
| Disease | 96.12±0.28 | **74.48±0.73** | 68.65±3.92 | 68.76±3.19 | 58.26±2.59 | 66.83±1.18 | 69.35±1.40 | 73.16±0.25 |
| Game | 90.11±0.31 | 64.91±1.07 | 52.97±6.99 | 66.82±3.05 | 51.38±4.03 | 53.43±2.10 | 58.69±0.92 | **74.34±0.75** |
| GPA | 71.07±1.48 | 34.39±0.84 | 44.14±3.00 | 45.72±0.44 | 50.60±0.64 | 42.47±2.13 | **50.97±0.17** | 41.90±0.56 |
| Student | 0.73±0.00 | 0.36±0.09 | 0.51±0.03 | -1.06±0.15 | 0.56±0.02 | -0.86±0.04 | **0.56±0.01** | 0.50±0.02 |
| **Avg Rank** | - | 4 | 4.5 | 4 | 4.3 | 5.5 | **2.3** | 3.3 |
| **N = 200** | | | | | | | | |
| Disease | 96.12±0.28 | 68.05±1.15 | 62.53±5.31 | 69.56±1.75 | 69.97±2.37 | 71.14±0.93 | 71.42±1.02 | **73.37±1.46** |
| Game | 90.11±0.31 | 64.73±6.83 | 55.00±0.75 | 72.15±1.45 | 49.36±2.42 | 64.08±1.33 | 65.07±2.17 | **85.32±0.51** |
| GPA | 71.07±1.48 | 37.51±1.29 | 43.25±0.90 | 45.75±0.44 | 47.37±0.94 | 41.27±2.75 | **48.80±0.58** | 42.32±0.67 |
| Student | 0.73±0.00 | 0.48±0.03 | 0.47±0.11 | 0.05±0.09 | 0.53±0.01 | 0.11±0.00 | **0.55±0.02** | 0.53±0.01 |
| **Avg Rank** | - | 5.3 | 5.5 | 4.3 | 3.8 | 5 | **1.8** | 2.3 |
| **N = 300** | | | | | | | | |
| Disease | 96.12±0.28 | **77.54±0.64** | 36.97±2.97 | 69.55±3.10 | 63.23±4.46 | 63.81±1.14 | 71.46±3.67 | 75.89±0.58 |
| Game | 90.11±0.31 | 78.80±0.97 | 61.16±6.76 | 66.01±1.25 | 56.64±1.64 | 51.63±1.22 | 65.03±1.18 | **86.15±0.69** |
| GPA | 71.07±1.48 | 38.52±0.69 | 39.44±1.13 | 44.72±0.98 | 43.71±0.74 | 46.75±1.42 | **47.10±0.26** | 44.84±0.63 |
| Student | 0.73±0.00 | 0.48±0.03 | 0.47±0.11 | 0.04±0.10 | 0.53±0.01 | 0.19±0.04 | **0.55±0.01** | 0.54±0.01 |
| **Avg Rank** | - | 3.5 | 5.8 | 4.5 | 5.3 | 4.8 | 2.3 | **2** |
| **N = 500** | | | | | | | | |
| Disease | 96.12±0.28 | 74.57±0.91 | 75.00±1.47 | 68.03±1.73 | 66.29±1.56 | 64.74±0.99 | 68.79±1.04 | **77.32±0.73** |
| Game | 90.11±0.31 | 80.55±0.73 | 63.73±6.80 | 80.08±1.50 | 70.13±4.25 | 73.50±2.38 | 78.44±0.25 | **88.33±0.31** |
| GPA | 71.07±1.48 | 36.60±0.96 | **46.48±1.01** | 45.28±0.63 | 40.00±0.90 | 38.44±1.08 | 42.55±0.48 | 42.68±1.06 |
| Student | 0.73±0.00 | 0.56±0.01 | 0.53±0.01 | -0.99±0.19 | 0.50±0.01 | -0.60±0.13 | **0.57±0.03** | 0.56±0.01 |
| **Avg Rank** | - | 3.5 | 3.5 | 4.3 | 5.5 | 6 | 3.3 | **1.8** |

Table 14: Comparison between Best Non-LLM baselines and LLM-based generators. All results are evaluated using a fixed **Linear/Logistic Regression** downstream model. All numbers are reported as **mean ± standard deviation**. Unseen datasets are highlighted in blue. The best result in each row is shown in **bold**, and the second-best result is underlined.

| Datasets | Real | CTGAN | DRL | TabSyn | GReaT | EPIC | CLLM | I+II |
|---|---|---|---|---|---|---|---|---|
| **N = 30** | | | | | | | | |
| Disease | 79.88±1.13 | 47.76±3.77 | 34.52±0.90 | 61.51±2.43 | 61.75±3.34 | 26.06±0.00 | 62.85±0.88 | **67.91±1.03** |
| Game | 77.62±0.63 | 27.69±3.32 | 19.79±3.99 | 20.45±1.05 | 32.89±11.78 | 35.47±3.15 | 37.67±0.95 | **41.87±1.47** |
| GPA | 47.95±1.88 | 10.75±3.46 | 10.75±3.46 | 32.15±5.34 | 39.68±1.14 | 38.46±1.50 | 27.94±1.33 | **43.21±0.40** |
| Student | 0.64±0.00 | -0.93±0.15 | -0.11±0.08 | 0.28±0.03 | 0.40±0.02 | 0.25±0.00 | 0.20±0.02 | **0.41±0.01** |
| **Avg Rank** | - | 5.8 | 6.3 | 4.3 | 2.8 | 4.3 | 3.5 | **1** |
| **N = 60** | | | | | | | | |
| Disease | 79.83±1.40 | 52.11±3.48 | 28.47±0.57 | 57.78±1.68 | 46.67±10.65 | 26.06±0.00 | 50.29±2.13 | **66.18±0.92** |
| Game | 77.62±0.64 | 30.39±2.33 | 29.48±4.18 | 49.52±18.14 | 31.71±10.26 | 36.83±0.95 | 42.47±0.93 | **55.87±1.25** |
| GPA | 48.28±1.81 | 18.35±1.17 | 22.39±2.69 | 21.85±1.95 | 35.41±1.96 | 32.21±1.32 | 16.82±1.60 | **39.56±0.11** |
| Student | 0.65±0.00 | -0.11±0.05 | -0.48±0.26 | 0.28±0.07 | 0.39±0.02 | 0.37±0.03 | -0.31±0.02 | **0.39±0.01** |
| **Avg Rank** | - | 5 | 6 | 3.3 | 3.3 | 4.3 | 5 | **1** |
| **N = 90** | | | | | | | | |
| Disease | 80.06±0.68 | 47.81±1.80 | 39.04±0.00 | 62.12±1.01 | 62.05±0.86 | 26.45±0.00 | 65.59±1.54 | **66.29±0.36** |
| Game | 77.69±0.89 | 30.37±4.44 | 25.28±3.77 | **69.21±0.59** | 34.64±13.34 | 59.37±1.06 | 14.52±2.30 | 56.15±2.33 |
| GPA | 48.32±1.64 | 15.39±4.21 | 16.02±2.00 | 36.62±2.87 | 32.56±4.86 | 30.89±1.51 | 25.55±0.75 | **44.77±0.51** |
| Student | 0.64±0.00 | -0.07±0.03 | -0.04±0.05 | 0.06±0.09 | 0.40±0.11 | 0.35±0.07 | -0.22±0.02 | **0.44±0.01** |
| **Avg Rank** | - | 5.8 | 5.8 | 2.5 | 3.3 | 4.8 | 4.5 | **1.5** |
| **N = 120** | | | | | | | | |
| Disease | 79.56±1.04 | 39.63±4.16 | 56.92±1.59 | 61.33±1.40 | 43.92±7.95 | 26.11±0.00 | 65.71±2.01 | **72.54±1.00** |
| Game | 77.55±0.34 | 26.56±2.60 | 35.73±5.78 | **56.33±1.45** | 41.82±13.59 | 46.40±1.22 | 47.09±2.09 | 49.85±0.50 |
| GPA | 50.55±1.30 | 17.81±2.14 | 14.25±3.64 | 40.79±3.42 | 27.45±11.06 | 31.89±1.51 | 38.50±0.29 | **44.48±0.77** |
| Student | 0.64±0.00 | 0.01±0.05 | 0.03±0.05 | 0.55±0.02 | 0.53±0.02 | **0.57±0.02** | -0.18±0.01 | 0.47±0.02 |
| **Avg Rank** | - | 6.3 | 5.5 | 2 | 4.5 | 4 | 3.8 | **2** |

# G Privacy Evaluation via Distance to Closest Record (DCR)

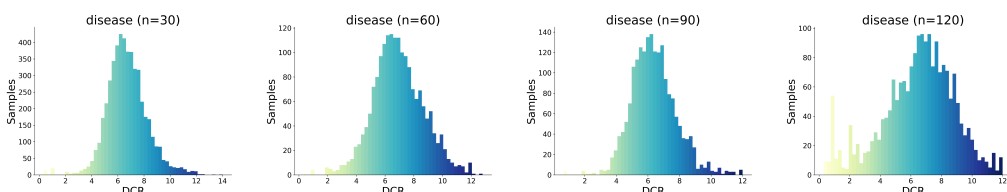

Figure 10: Distance to closest record (DCR) distributions for the DISEASE dataset, comparing synthetic data generated by **ReFine** to the original train set.

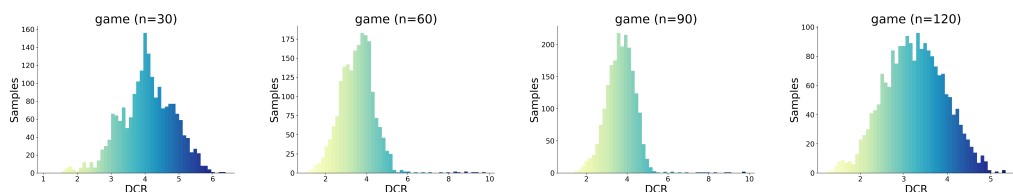

Figure 11: Distance to closest record (DCR) distributions for the GAME dataset, comparing synthetic data generated by **ReFine** to the original train set.

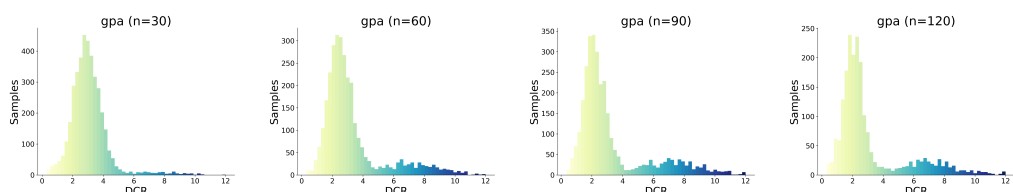

Figure 12: Distance to closest record (DCR) distributions for the DISEASE dataset, comparing synthetic data generated by **ReFine** to the original train set.

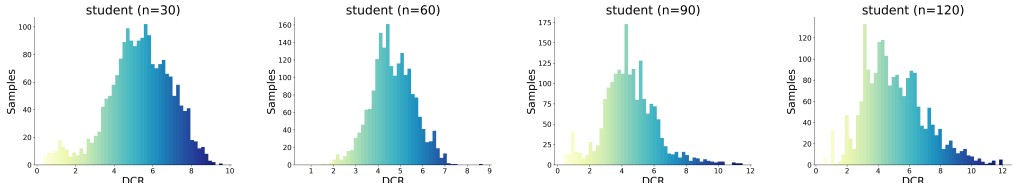

Figure 13: Distance to closest record (DCR) distributions for the DISEASE dataset, comparing synthetic data generated by **ReFine** to the original train set.

To assess whether synthetic samples inadvertently reproduce training records, we follow prior work on tabular data synthesis and report the *Distance to Closest Record (DCR)*. For each generated instance $s_{\text{gen}}$, DCR computes its distance to the nearest neighbour in the original training set $T_{\text{train}}$:

$$\text{DCR}(s_{\text{gen}}) = \min_{x \in T_{\text{train}}} d(s_{\text{gen}}, x), \tag{6}$$

where the mixed-type distance $d(\cdot, \cdot)$ is defined as an $L_1$ distance on numerical features and a $0/1$ contribution for categorical features (0 if equal, 1 otherwise). Small DCR values, particularly those close to 0, would indicate potential memorization or replication of training instances.

## G.1 DCR Results and Analysis

Figures 10–13 report DCR histograms for four unseen datasets (Disease, Game, GPA, Student) under training sizes $N \in \{30, 60, 90, 120\}$. Each histogram shows the distance between a generated sample and its closest training instance, computed using **DCR**.

Across all datasets and all low-data regimes, two consistent observations emerge:

**(i) No evidence of memorization.** Even at the smallest data setting ($N = 30$), the DCR mass remains well separated from zero: we do not observe spikes at distance 0 nor anomalous concentration near 0. This indicates that ReFine does not reproduce training records verbatim, despite the extreme data scarcity.

**(ii) Stable privacy behaviour as $N$ increases.** As $N$ grows from 30 to 120, the DCR distribution shifts slightly toward smaller values—reflecting the fact that more structure becomes identifiable—but it remains broad and non-degenerate. Importantly, the distributions never collapse toward zero, showing that increased data availability does not lead to sample-level leakage.

In summary, ReFine maintains strong privacy behaviour across all low-data settings: it improves downstream learning without memorizing or leaking individual training records, even when only 30 examples are available.

# H  PROMPT FOR COMPONENT I

## Generalization Prompt

You are an intelligent assistant responsible for transforming nested if–else classification logic into generalized, structured rules for data generation. These rules involve a target label called Diagnosis (values 0 or 1) determined by thresholds across multiple input features such as MMSE, FunctionalAssessment, ADL, SystolicBP, SleepQuality, etc.

Your task consists of three main steps:

### Step 1: Parse If–Else Logic into Flat Rule Format

Given one or more nested if–else decision structures (represented by placeholders), flatten them into a set of human–readable rule statements. Each statement must describe the conditions that lead to a specific Diagnosis assignment.

#### Input:
{if_else_logic}

Repeat this process for all provided if–else blocks. The output should be a set of conditions grouped by Diagnosis.

### Step 2: Merge Similar Diagnosis Rules Across Samples

Merge rules belonging to the same Diagnosis across multiple samples.

#### Merging Logic:
– Identify all rules corresponding to a given Diagnosis.
– For each feature (e.g., MMSE), extract the relevant conditions across all rules.
– Instead of strictly choosing the second smallest or second largest value, focus on:

1. **Maintaining differentiation** between adjacent Diagnosis, ensuring that boundaries between different Diagnosis are clearly defined and preserved.
2. **Maximizing generalization**, meaning that overlapping condition ranges should be merged in such a way that it still respects the distinctions while covering the broadest possible range.

### Step 3: Output Final Rule Set

Produce a **final set of generalized rules**, but **only one rule per Diagnosis** (0 or 1). The rules should strictly incorporate only mention in the **Key Features with Importance**, **strictly** following this format:

– If Diagnosis = 0, then [CONDITION1] and [CONDITION2] ...
– If Diagnosis = 1, then [CONDITION1] and [CONDITION2] ...

Where each condition is a feature threshold based on the merging strategy above, using only the **Key Features with Importance**. Each rule should focus on these features, and you should only have one rule for each Diagnosis.

Make sure your output:
– Uses readable formatting
– Maintains clear zones between different Diagnosises, – Maintains clear **while allowing overlap where necessary**
– Uses inclusive and exclusive operators appropriately (i.e., >, >=, <, <=)
– Properly incorporates feature importance to prioritize significant features
– Avoids overly specific or restrictive rules

Figure 14: **Rule Generalization Prompt.** We prune and consolidate decision paths across top-performing trees, retaining only their core (i.e., high-support, low-depth) branches that reflect stable feature–label dependencies. This yields generalized *if–then* rules that extend beyond specific training instances. Treated as conditional templates with the label as premise, these rules enable inverse reasoning and guide sample generation toward broader yet distributionally faithful regions of the data space.

---

**Denoising Prompt**

---

You are an intelligent assistant tasked with applying a self–consistency methodology to combine multiple sets of regression–derived interval–based rules.

You have received five independently generated rule–sets from multiple executions of previous analysis steps. Each rule–set consists of interval–based rules predicting continuous–values (e.g., MMSE) with specific conditions. Your goal is to integrate these five sets of rules into one consistent, summarized, and robust rule set.

### Input: (Insert your five independently generated rule–sets below)

{Rule Set 1}, ... ,{Rule Set 5}

### Step 1: Identify Consistent and Robust Rule Patterns

For each predicted feature (e.g., Diagnosis), examine all five rule–sets and perform the following:

– Identify stable, frequently occurring conditions and intervals across rule–sets.
– For intervals appearing consistently (at least 3 out of 5 times), retain them as robust and representative intervals.
– If intervals differ slightly across sets, choose interval bounds that reflect the majority consensus, using [mean(min_values), mean(max_values)].

### Step 2: Generate a Single Consistent Interval–based Rule Set (Self–consistency)

Produce a single final integrated rule–set incorporating the majority consensus intervals clearly. Format rules as follows:

– If [Predicted Feature] = [label], then [CONDITION1] and [CONDITION2]...

Ensure the final output:
– Clearly represents stable intervals identified across multiple rule–sets.
– Uses inclusive/exclusive operators properly (>, ≥, <, ≤).
– Avoids overly broad intervals, prioritizing clarity and consistency.

---

Figure 15: **Rule Denoising Prompt.** To mitigate variance introduced by symbolic extraction and decoding noise (He et al., 2024), we adopt self-consistency strategies commonly used in reasoning tasks (Wang et al., 2023; Lewkowycz et al., 2022). Multiple generations are sampled using different random seeds, and only the most frequently occurring rules are retained. This procedure stabilizes the extracted rule set and ensures its reliability for downstream generation.

$$\text{RMSE}_i = \sqrt{\frac{1}{T} \sum_{t=1}^{T} (\hat{y}_{i,t} - y_i)^2}.$$

