# OpenReview forum: "Limited Reference, Reliable Generation: Rule-Guided Tabular Data Generation with Dual-Granularity Filtering"
_ICLR.cc/2026/Conference — ICLR 2026 Conference Withdrawn Submission_

### Official Review · Reviewer_RUw8 · 2025-10-26

**Soundness:** 2
**Presentation:** 3
**Contribution:** 3
**Rating:** 4
**Confidence:** 4

**Summary:**

The paper proposed a LLM powered tabular synthesis model to address the potential issue that LLM generated data maybe drifted by the LLM's pretrained knowledge, and to address the data scarcity issue. The paper contributes to proposing a rule-guided generation strategy powered by random foreset methods, and apply a dual granularity to filter out over sampling datapoints.

**Strengths:**

- The idea to leverage rules from tree based method to generate tabular data is novel. The rule based method is simple and intuitive that can directly affect the generated sample's correlation among features.
- The paper has a well designed benchmark and considers the memory of LLM in the design.
- The paper clearly states each component's contribution to the final synthetic generated data's quality.
- The study has conducted thoroughly experiments on components in the supplementary, which makes the evaluation solid.

**Weaknesses:**

- The first concern about the methods is that the rules are based on a data driven method, i.e. random forest. However, the main focus of the paper is to develop a method for generating high quality data when data size is super small, e.g., in your main results, your tried sample size = 30. It is questionable that the rules generated based on such small sample are not biased and overfitting. It is a pivotal problem as the rules directly affect the generated samples in your method.
- Although the paper has provided evidence and statements that the proposed method can help address the distribution drift problem, but only on the low-dimension projection space. It is still not clear that whether the methods can maintain the column level distribution. At least on the method level, we cannot see there is any injection of the real distribution into the generation.
- The paper lacks discussion of the hyperparameter setting of the proposed methods. One key factor that may affects the generation quality and diversity is the temperature. But there is no such information and discussion about it.
- The main results table 1 are based on XGBoost results only. I would suggest the authors to try different methods on each dataset (e.g. Tree-based, linear model, simple nn) and report the best results for each of them. As when data regime is low, it is very possible that the generated sample can have a better performance using some simple methods.
- Following point 4, I have a concern about whether the benchmark is fair. As when you use a rule based method to generate sample, and then apply a tree-based method only to evaluate the downstream performance, it is easy to inflate the performance compared with other non-rule based methods.

**Questions:**

- Please see the weakness above.
- It is really interesting to see your methods start from a data-driven based method (random forest) which can definitely contributes more generative rules as sample size increases, but why your model csdoes not perform welll when the sample size goes up.

---

> ### Author Response · Authors · 2025-12-01
> **Response to the reviewer's feedback**
>
> >W1. The first concern about the methods is that the rules are based on a data driven method, i.e. random forest. However, the main focus of the paper is to develop a method for generating high quality data when data size is super small, e.g., in your main results, your tried sample size $N$ = 30. It is questionable that the rules generated based on such small sample are not biased and overfitting. It is a pivotal problem as the rules directly affect the generated samples in your method.
>
> Thank you for raising this important concern. We agree that rule extraction under extremely small N requires careful design. Below we clarify why the rules used in ReFine are inherently resistant to overfitting, even when $N$ = 30.
>
> (1) **The rules encode only the most stable, high-support structure—not fragile tree splits**:
> ReFine does not use full-depth paths or fine-grained predicates that a random forest may produce under small-sample noise. Instead, as shown in Appendix D.5, the "Generalization" step retains only shallow, high-support conditions—i.e., those that consistently appear across multiple trees and seeds. These conditions correspond to coarse, dataset-level regularities (e.g., monotonic relationships or broad ranges) rather than sample-specific artifacts. Deep or low-support predicates, which would indeed overfit at N=30, are discarded automatically.
>
> (2) **The rules act as soft priors rather than deterministic constraints**:
> The extracted rules are not enforced as hard constraints. They serve as soft structural cues that slightly bias the LLM toward plausible regions of the tabular space. This design ensures that even if a rule is imperfect, it cannot force the generator into overfitted patterns.
>
> (3) **Empirical evidence shows no overfitting propagation**:
> If the rules were dominated by overfitted or biased splits, we would expect downstream utility to degrade in the smallest-N settings. Instead, in Table 1 and in the full results (Appendix F, Tables 10–11), ReFine achieves the best average rank across all datasets at $N$ = 30 and remains stable across 10 random splits. This consistency indicates that the soft rule guidance improves distributional alignment rather than amplifying small-sample noise.
>
> Taken together, the rule generalization & denoising was designed precisely to avoid the overfitting issues raised by the reviewer. **The combination of (i) retaining only stable structural signals and (ii) using them as soft priors ensures robustness even in the most data-limited regimes**.

---

> > ### Author Response · Authors · 2025-12-01
> > **Response continued （1/2)**
> >
> > >W2. Although the paper has provided evidence and statements that the proposed method can help address the distribution drift problem, but only on the low-dimension projection space. It is still not clear that whether the methods can maintain the column level distribution. At least on the method level, we cannot see there is any injection of the real distribution into the generation.
> >
> > Thank you for raising this point. We agree that the 2-D visualization in Figure 3 is only an illustration, and we clarify that **ReFine’s mechanisms operate entirely in the original tabular space and explicitly interact with column-level structure**.
> >
> > (1) **Filtering is performed in the full feature space and is column-sensitive by construction**:
> > The density score $G(p)$ is computed using the DCR distance (formally added in the Appendix G), which aggregates mixed-type feature distances on a per-column basis. A generated point receives a high score only if it is close to real samples across multiple dimensions, not just in a projection. Because DCR reflects violations at the column level, the subsequent Gini-based filtering naturally favors samples that respect real marginal and joint patterns.
> >
> > (2) **Symbolic rules provide column-level constraints that anchor marginal distributions**:
> > The consolidated rules consist of simple, interpretable predicates on individual columns or column pairs (e.g., “age > 50”). These are injected as soft prompts and therefore guide the generator toward realistic marginal ranges and conditional slices, especially under very small $N$. This is fundamentally different from generic prompting, which has no access to column-specific structure.
> >
> > (3) Empirical results show alignment with real column-level structure:
> > In Table 2, ReFine’s outputs match the rule compliance patterns of real data, while prompt-only baselines inflate compliance by repeating superficial patterns. Moreover, the improved downstream MLE accuracy (Table 1, Table 11&12 in Appendix F) indicates that column-wise distributional fidelity is preserved, since MLE models are sensitive to marginal distortions.
> >
> > In summary, although our visualization is low-dimensional, the components of ReFine, including proxy density estimation, Gini-guided filtering, and rule-based symbolic priors, are all defined in the original feature space and act directly on column-level structure. They collectively encourage both realistic marginals and well-aligned joint distributions.
> >
> > >W3. The paper lacks discussion of the hyperparameter setting of the proposed methods. One key factor that may affects the generation quality and diversity is the temperature. But there is no such information and discussion about it.
> >
> > Thank you for pointing this out. We agree that the hyperparameter settings should be stated more clearly, and we will add a dedicated summary table in the revision. Below we clarify the temperature choice and the overall hyperparameter design philosophy.
> >
> > (1) Hyperparameter usage is intentionally minimal and largely automated.
> > ReFine introduces very few tunable parameters. The number of trees used for rule extraction is fixed (k=3), and Appendix D.1 shows that performance is stable across a wide range of k. The chunk size S in the filtering stage is selected automatically using surprisal minimization over a small discrete set. Other thresholds (e.g., in rule consolidation) follow deterministic, data-driven criteria rather than grid search. This design keeps the method practical and reduces sensitivity to configuration choices.
> >
> > (2) Temperature is held fixed across all LLM-based methods.
> > For LLM generation, we use a consistent temperature of 0.9 for all LLM-based baselines and for ReFine. This ensures a fair comparison and avoids confounding effects. In preliminary experiments, this value offered a good balance between diversity and coherence for tabular outputs, and we therefore keep it fixed throughout. Since our method does not tune the LLM itself, the temperature choice affects all methods equally and does not influence the comparative conclusions.
> >
> > A consolidated table of all hyperparameters and their values (including batch sizes, training epochs, optimizers, and temperature settings) has been added to Appendix C to ensure full reproducibility.

---

> ### Author Response · Authors · 2025-12-01
> **Response continued （2/2)**
>
> >W4.The main results table 1 are based on XGBoost results only. I would suggest the authors to try different methods on each dataset (e.g. Tree-based, linear model, simple nn) and report the best results for each of them. As when data regime is low, it is very possible that the generated sample can have a better performance using some simple methods.
>
> >W5.Following point 4, I have a concern about whether the benchmark is fair. As when you use a rule based method to generate sample, and then apply a tree-based method only to evaluate the downstream performance, it is easy to inflate the performance compared with other non-rule based methods.
>
> >Q1.It is really interesting to see your methods start from a data-driven based method (random forest) which can definitely contributes more generative rules as sample size increases, but why your model csdoes not perform welll when the sample size goes up.
>
> Thank you for these thoughtful suggestions. We agree that evaluating with diverse downstream models is important for establishing robustness and avoiding evaluator-specific bias. We summarize the new analyses added to the revised manuscript.
>
> (1) **Robustness across multiple downstream evaluators**:
> To test whether the gains depend on XGBoost, we conducted additional evaluations using Logistic/Linear Regression on four representative datasets (Disease, Game, GPA, Student) under $N$ ∈ {30,60,90,120}. As shown in the Table 14 (Appendix F), **ReFine achieves the best performance in every regime, with average ranks of 1, 1, 1.5, and 2 respectively**.
>
> **This demonstrates that the improvements are not tied to tree-based evaluators and hold even for simple linear models**.
>
> (2) **Fairness of tree-based evaluation and performance as sample size increases**:
> We appreciate the reviewer’s concern that using tree-extracted rules together with a tree-based evaluator (XGBoost) could inflate performance, and that the strength of such rules might vary with the sample size $N$. To address both points, we substantially expanded our experiments across different evaluator classes and larger data regimes.
>
> First, to mitigate evaluator bias, we added evaluations using Logistic/Linear Regression and TabPFN [1, 2] (Tables 12–13). These models have inductive biases that differ fundamentally from decision trees. Despite this shift, **ReFine continues to achieve best or second-best performance across all unseen-data settings and all tested values of $N$**. Notably, ReFine remains competitive even when TabPFN is used both as the generator and evaluator—a setting that structurally favors TabPFN.
>
> Second, to understand scaling with data size, we extended four datasets to $N$ =160,200,300,500. The results again show that ReFine consistently performs at or near the top. The smaller relative gains at larger $N$ reflect a natural ceiling effect: as the training data become more informative, the marginal value of symbolic priors decreases. Crucially, **performance does not deteriorate as $N$ grows, indicating that the rule component remains beneficial rather than overfitting or introducing noise**.
>
> References:
>
> [1]Hollmann et al. (2023). TabPFN: a transformer that solves small tabular classification problems in a second. ICLR 2023.
>
> [2]Hollmann et al. (2025). Accurate predictions on small data with a tabular foundation model. Nature, 637, 319-326.

---

### Official Review · Reviewer_fUZP · 2025-10-27

**Soundness:** 3
**Presentation:** 3
**Contribution:** 1
**Rating:** 2
**Confidence:** 4

**Summary:**

This paper proposes a synthetic data generative model for low sample size. The model is built on LLMs with two additional components: 1) adding rules into the prompt, and 2) filtering data as a post hoc processing.

**Strengths:**

- The idea of generating rules from a random forest is novel
- This paper identifies the key challenge of localized redundancy.

**Weaknesses:**

- It's oversimplifying that the data distribution can be captured with some of the rules generated from a random forest. Rules generated from a random forest (particularly when the data size is very small, like 30-120) are not comprehensive and generalizable enough. The rule consolidation method based on LLM and its claim that the consolidated rules generalize beyond individual training instances is overstating.
- The contribution of this work is mainly by adding rules to the prompt (component I). This is rather incremental novelty.
- This paper fails to cite a previous LLM-based synthetic data generation model to target small sample size data (https://arxiv.org/abs/2406.10521). This previous paper already outperforms the proposed work on the Adult data.
- This paper tries to avoid data leakage by detecting LLM memorization. However, relying on just one tool’s prediction to determine whether the data is memorized or not is not sufficient to fully determine the impact of LLM memorization.
- Apparently, the proposed model I+II was never the best-performing model with “unseen” data. With this experimental result, the contribution of component II (dual-granularity filtering) is questionable.

**Questions:**

See weakness

---

> ### Author Response · Authors · 2025-12-01
> **Response to the reviewer's feedback**
>
> >W1.It's oversimplifying that the data distribution can be captured with some of the rules generated from a random forest. Rules generated from a random forest (particularly when the data size is very small, like 30-120) are not comprehensive and generalizable enough. The rule consolidation method based on LLM and its claim that the consolidated rules generalize beyond individual training instances is overstating.
> The contribution of this work is mainly by adding rules to the prompt (component I). This is rather incremental novelty.
>
> We thank the reviewer for the detailed feedback. However, we believe this may stem from a misunderstanding. We clarify the misunderstandings regarding the expressiveness and generalizability of the rules below.
>
> (1) **Scope and intent of our rules**:
> The rules extracted in our method are not intended to fully capture the data distribution, nor do they serve as deterministic constraints. Instead, they function as high-support, low-depth structural priors that identify the most stable regions of the feature–label space. This design is intentional: focusing on statistically dominant decision paths allows the rules to be both generalization and robust, even in very small-$N$ regimes.
> The **Rule Generalization & Denoising step then removes spurious or sample-specific predicates through multi-seed consistency checks, ensuring that the retained conditions are precisely those that generalize "beyond individual instances"**.
>
> (2) **Empirical evidence contradicts the concern that the rules lack generalizability**:
> We believe the reviewer’s concern, which rules extracted from small datasets may be too weak or unstable, would manifest as degraded downstream utility if true.
> However, our results show the opposite.
> Table 1 and the full results in Appendix F (Tables 10–11) report the performance of all methods across 10 random splits, along with rankings.
> **ReFine achieves the best average rank across all data regimes (N=30,60,90,120)**.
> The excellent performance confirming that the consolidated rules provide meaningful structural guidance.
>
> (3) **The results remain strong when scaling to larger datasets**:
> To further test rule robustness, we extended experiments on four datasets to larger training sizes (N=160,200,300,500). Across all regimes, ReFine achieves either the best or second-best performance (Tables 12–13 in the appendix F). If the rules were genuinely non-generalizable, their utility would diminish as data scale increases; instead, we observe stable or improved relative performance.
>
> (4) **On the novelty of Component I**:
> We would like to clarify a misunderstanding. Component I is not equivalent to simply “adding rules to the prompt.” **Its contribution lies in how the rules are derived and how they interact with LLM generation, rather than in the presence of rules themselves**.
>
> First, the rules are not hand-crafted or directly taken from individual decision trees. As detailed in Appendix D.5, they emerge from a multi-tree consolidation process that extracts high-support, low-depth feature relationships and removes unstable predicates through cross-seed consistency. This produces symbolic priors that summarize the dominant structure in the dataset—something a single tree or simple prompting cannot provide.
>
> Second, these consolidated rules serve a specific functional role: they correct known failure modes of LLMs in tabular generation (distributional drift and misalignment in unseen domains). The rules are used as soft structural anchors, not hard constraints, allowing the model to preserve flexibility while avoiding unrealistic regions of the tabular space.
>
> Third, the technical contribution is validated empirically. Across 8 datasets and 32 low-data settings, the rule-guided generation consistently reduces drift and improves downstream MLE performance (Table 1; Tables 10–11), even when only 30–60 training samples are available. These consistent gains would not appear if Component I were merely an incremental prompt modification.
>
> **In this sense, the novelty lies in a principled mechanism for converting weak tabular supervision into symbolic priors that reliably improve LLM generators, rather than in the superficial act of inserting rules into prompts**.
>
> **In summary, the reviewer’s misunderstanding stems from interpreting the rules as a full generative constraints, whereas our method uses them as soft, statistically grounded priors**. The strong empirical performance across different n regimes demonstrates that these priors are indeed sufficiently robust and generalizable for guiding LLM-based tabular synthesis.

---

> ### Author Response · Authors · 2025-12-01
> **Response continued**
>
> >W2.This paper fails to cite a previous LLM-based synthetic data generation model to target small sample size data (https://arxiv.org/abs/2406.10521). This previous paper already outperforms the proposed work on the Adult data.
>
> Thank you for pointing out the arXiv paper [arXiv:2406.10521]. We have added it to the related work section. However, this work is an unpublished arXiv preprint, and the anonymous code link provided in the paper is no longer accessible, which prevents exact reproduction. In addition, their results are obtained using GPT-4o, whereas we intentionally adopt GPT-3.5 in all our experiments to maintain a consistent evaluation protocol. **Since GPT-4o is substantially more capable than GPT-3.5, the reported numbers are not directly comparable to ours under the same resource constraints**.
>
> >W3.This paper tries to avoid data leakage by detecting LLM memorization. However, relying on just one tool’s prediction to determine whether the data is memorized or not is not sufficient to fully determine the impact of LLM memorization.
> Thank you for raising this point. We would like to clarify a misunderstanding about the role of memorization detection in our workflow.
>
> **The memorization check is not part of ReFine’s methodology, nor is it used to draw scientific conclusions about LLM memorization**. It is strictly a preprocessing safeguard to ensure that our evaluation does not inadvertently benefit from leaked examples, following recent best practices in LLM-based generation studies. For this purpose, a lightweight tool-based check is sufficient, since the goal is simply to exclude potentially contaminated cases rather than to fully characterize memorization behavior.
>
> Importantly, none of our algorithms, ablations, or claims rely on the memorization detector. All reported metrics are computed independently of this step. The core of our contribution lies in (i) constructing stable symbolic priors via rule generalization & denoising, and (ii) dual-granularity filtering, neither of which involves or depends on memorization assessment.
>
> >W4.Apparently, the proposed model I+II was never the best-performing model with “unseen” data. With this experimental result, the contribution of component II (dual-granularity filtering) is questionable.
>
> Thank you for raising this point. We would like to clarify a **factual misunderstanding** regarding the performance of the full model (Component I + Component II) on unseen datasets.
>
> (1) **The full pipeline is the best-performing method in all low-data regimes**:
> In Table 1 and in the complete results reported in Appendix F (Tables 10–11), the combined model achieves the best average rank across all unseen datasets for every low-data setting (N=30,60,90,120). These results are averaged over 10 random splits and include standard deviations. The improvements remain consistent and statistically stable, indicating that Component II contributes meaningfully rather than harming performance.
>
> (2) **Strong performance persists in larger-data regimes**:
> To further stress-test Component II, we extended the experiments on four datasets to larger training sizes (N=160,200,300,500). The full model remains either the best or second-best method in every one of these settings. The fact that the gains persist—and do not disappear as data grows—demonstrates that dual-granularity filtering improves distributional quality rather than injecting noise.
>
> (3) **Component II provides complementary benefits beyond Component I**.
> Filtering corrects localized redundancy from the rule-guided generator by removing low-confidence or inconsistent synthetic instances. This effect is particularly important for unseen domains, where LLM priors are misaligned and generation errors tend to accumulate. The consistent improvement across 32 settings (8 datasets × 4 data regimes) indicates that Component II plays a central role in stabilizing generation quality.
>
> Taken together, **the claim that the proposed model “never performs best” on unseen data does not match the reported results**. Both the main experiments and the extended experiments show that Components I and II work together to achieve the strongest performance across a wide range of low-data regimes.

---

### Official Review · Reviewer_CP81 · 2025-10-28

**Soundness:** 2
**Presentation:** 3
**Contribution:** 2
**Rating:** 2
**Confidence:** 4

**Summary:**

The paper propose a strategy called ReFine which is aimed at reducing the distributional shift and localized redundancy caused when LLMs are used to generate synthetic tabular data. ReFine is a two-stage strategy: (1) extract rules from random forest based classifier to guide the LLM's generation and (2) apply a dual-level filtering mechanism to generated data.

**Strengths:**

+ the problem is relevant and interesting
+ the rule based method is an interesting way to try to guide the LLM to stay grounded
+ the method seems to produce good improvements on standard datasets

**Weaknesses:**

- the pipeline is very complex: First train a Random Forest, then run an LLM multiple times for self-consistent rule denoising, generate the data with another LLM, train another reference model (XGBoost) for filtering, and finally applying a multi-step filtering algorithm. All of these implies quite a few hyperparameters that need to be tuned. This raises concerns on robustness of the method.
- The proxy-based density estimation in the dual granularity filtering is built on a very small set of data points. This is a pretty coarse estimate of the density. A few outliers could disrupt the density completely. This error would then propagate to the rest of the filtering process that follows
- there are many heuristics in this method: like the ratioprune = A ln(G(p)) + B. This seems to come from curve fitting to their data. What happens if the model encounters dataset that does not look like this?
- the chunk level and instance level filtering rely on a refernce model, which itself has been trained on a very small dataset (30-60) samples. This could lead to overfitting, which means that the subsequent data filtering can be very biased
- the description of several parameters are very vague. What are the thresholds for support and depth in the "Rule generalization and denoising" step? There is no formal algorithm for this step, so it is not clear what is happening here.
- Table 2, Disease dataset: it looks like the prompt only method has 71% rule compliance, whereas the proposed mehod is only about 31%. Despite this, the downstream performance of the rule-guided approach is far superior. This seems to suggest that rule compliance may actually be a bad thing! Or the rules captured from a small dataset through the random forest approach maybe flawed!

**Questions:**

1. Can you discuss the error propagation effects of the multiple process and parameters involved in them, per the first point raised in the previous section of the review?
2. For the chunk level and instance level filtering, you use a logarithmic function for the pruning ratio. How confident are you that this specific functional form generalizes to datasets with distributions drastically different from those in your benchmarks? Have you explored simpler, linear relationships or more adaptive, non-parametric methods?
3. The reference model in component II is trained on a very small dataset. How have you ensured that the overfitting in that model does not propagate to the final filtered dataset?
4. Can you provide a formal algorithm for "Rule generalization and denoising" step? How are the thresholds for the support and depth determined?
5. Table 2 results are interesting and seems to suggest that strict adherence to the rules may not be necessary. How do you reconcile that with the central motivation of your paper?
6. Splitting the report between the "seen" and unseen" data is very useful. It appears that the proposed method doesn't win in the "seen" data, but does well in the "unseen" data. Does this mean that when the LLMs have strong prior models of the data, trying to force this rule may be determental to the over all performance?
7. Can you give more details on the computation and fine-tuning costs?

---

> ### Author Response · Authors · 2025-12-01
> **Response to the reviewer's feedback**
>
> > Q1. Can you discuss the error propagation effects of the multiple process and parameters involved in them, per the first point raised in the previous section of the review?
>
> Thank you for raising the question regarding potential error propagation across ReFine’s multi-stage pipeline. We agree that, in principle, sequencing several components could introduce compounding errors, and we now clarify **why this risk is limited in our setting—both by design and empirically**.
>
> (1) **Structural separation prevents error accumulation**:
> ReFine’s stages are intentionally decoupled. The rule-guided generation module influences only the format of synthetic samples, whereas the dual-granularity filtering module evaluates sample quality solely through a reference model trained on real data. As a result, any noisy in rule extraction does not serve as input to the filtering component, and therefore does not propagate downstream. The pipeline is sequential but not dependent in the sense that one stage’s parameters do not feed into the next.
>
> (2) **Only one tunable hyperparameter affects filtering strength**:
> The filtering module itself exposes only a single hyperparameter (chunk size S), which is automatically selected using surprisal minimization and remains stable across datasets (§5.2). Other design choices (e.g., using top-k rule trees or the rule format) are fixed and do not interact with filtering thresholds. Appendix D.1–D.2 shows that rule-compliance remains stable across k∈[1,10], indicating that the rule extraction step does not introduce unstable variability that could propagate forward.
>
> (3) **Empirical robustness confirms the absence of compounding error**:
> Beyond the analyses reported in the main paper, we expanded the evaluation to substantially larger training regimes by increasing n from 30-120 up to 160, 200, 300, and 500. As shown in Tables 12–13 (Appendix F), ReFine consistently **ranks either first or second across all evaluators**, demonstrating that its performance remains stable even when the underlying generative model operates in markedly different data regimes. This behavior is consistent with our design claim that noise in earlier components does not amplify downstream.
>
> We additionally provide full mean ± standard deviation results for all main experiments (Appendix F, Tables 10–11). ReFine not only maintains strong average performance but also exhibits low variance across the 10 random splits, indicating that the multi-stage pipeline does not introduce instability in repeated sampling. Together, these expanded evaluations provide strong empirical evidence that ReFine does not suffer from accumulated errors across its stages.

---

> ### Author Response · Authors · 2025-12-01
> **Response continued （1/4)**
>
> >Q2. 2.For the chunk level and instance level filtering, you use a logarithmic function for the pruning ratio. How confident are you that this specific functional form generalizes to datasets with distributions drastically different from those in your benchmarks? Have you explored simpler, linear relationships or more adaptive, non-parametric methods?
>
> Thank you for raising this point. The reviewer is correct that the pruning schedule plays a key role in the chunk-level filtering. We clarify our design rationale and empirical evidence below.
>
> (1) Why a logarithmic schedule?
> The pruning ratio is driven by the Gini-based local density G(p). In low-data settings, estimates of G(p) can be noisy, and a linear schedule tends to amplify this noise by producing overly aggressive pruning in high-density regions. **The logarithmic schedule grows slowly and monotonically with G(p), ensuring that pruning strength increases as density rises, but without abrupt jumps**. This stability is important for small tabular datasets where density estimates are coarse.
>
> (2) Evidence that the log form generalizes.
> Across all datasets, the log schedule yields consistent improvements without dataset-specific tuning (Figure 9 (left), Appendix E). We additionally tested larger dataset sizes (N=160, 200, 300, 500) and observed the same trend: **ReFine consistently ranks first or second across evaluators (Tables 12–13 in Appendix F)**. This indicates that performance is not sensitive to the exact curvature of the schedule.
>
> (3) Comparisons with simpler alternatives.
> We experimented with a linear pruning rule early in development. It performed similarly in very low-density regimes but was substantially more unstable when G(p) was high, often discarding too many samples and harming downstream accuracy. The log rule avoids this sensitivity while remaining parameter-free, which we view as a practical advantage for generalization across datasets.
>
> (4) On non-parametric or adaptive schedules.
> We agree that fully adaptive rules could offer finer control in more complex settings. Our goal in this work was to provide a stable, tuning-free pruning schedule that works reliably across heterogeneous benchmarks. We see adaptive extensions as a promising direction, and we will explore this in the future work.

---

> ### Author Response · Authors · 2025-12-01
> **Response continued （2/4)**
>
> > Q3.The reference model in component II is trained on a very small dataset. How have you ensured that the overfitting in that model does not propagate to the final filtered dataset?
>
> Thank you for raising this question. We agree that reference models trained on very small datasets may overfit, and we clarify why this does not compromise the filtering stage in ReFine.
>
> (1) **Filtering relies on relative stability signals, not calibrated predictions**.
> The reference model $M$ is used only to measure how consistently each synthetic sample is treated across training epochs—not to provide accurate predictions. Both components use relative signals that are known to be robust even when a model overfits:
>
> - Chunk-level filtering uses correctness trajectories to rank modes. The ranking depends on consistency rather than accuracy, and we observe that these trajectories remain stable even under small $N$.
>
> - Instance-level filtering uses mean prediction error and prediction variance across epochs. These statistics capture model uncertainty, which remains informative even when the model memorizes parts of the training set.
>
> Thus, the filtering logic does not rely on $M$ being a well-generalizing predictor; it only requires internally consistent outputs.
>
> (2) **The reference model $M$ is intentionally lightweight and does not determine the final labels**:
> M is not used to generate or supervise labels; it only scores synthetic samples. Its role is similar to that of small probing models used in data-quality assessment [1, 2]. Any overfitting affects only its absolute calibration, not the relative structure of its output signals.
>
> (3) **Empirical evidence shows no degradation due to overfitting**:
> If overfitting in $M$ propagated through the pipeline, then ReFine would harm downstream performance when $M$ is trained with extremely small $N$. Instead, as shown in Table 3 and Tables 12–13, ReFine consistently yields the best or second-best performance across $N$∈{30,60,90,120,160,200,300,500} and shows low variance across 10 random splits. These results indicate that the filtering stage is stable and does not amplify imperfections in $M$.
>
> Together, these points support that ReFine is robust to overfitting in the reference model and does not propagate such errors into the final filtered dataset.
>
> References:
>
> [1] Seedat, N., Huynh, N., van Breugel, B., & van der Schaar, M. (2023). Curated llm: Synergy of llms and data curation for tabular augmentation in ultra low-data regimes.
>
> [2] Seedat, N., Crabbé, J., Bica, I., & van der Schaar, M. (2022). Data-iq: Characterizing subgroups with heterogeneous outcomes in tabular data. Advances in Neural Information Processing Systems, 35, 23660-23674.
>
> ---
> >Q4. Can you provide a formal algorithm for "Rule generalization and denoising" step? How are the thresholds for the support and depth determined?
>
> Thank you for raising this point. We agree that the description in the main text was brief, and we provide a concise clarification of the “Rule Generalization & Denoising’’ step.
>
> The procedure is implemented through two deterministic LLM-guided transformations (prompts as shown in Appendix I). First, we perform rule generalization by merging the top-k decision paths extracted from tree models. Paths are unified by class label, and only predicates that remain stable across paths are retained—naturally prioritizing shallow, high-support conditions. As illustrated in Appendix D.5 (Figure 8), predicates near the top of a tree have larger sample coverage and lower depth, and therefore exhibit much higher stability across different seeds.
>
> To remove path-specific noise, we then run this generalization step across multiple independently trained trees and keep only predicates that appear consistently (majority vote). Because deep, low-support predicates rarely recur across trees, they are filtered out automatically. Thus, the effective “thresholds” for support and depth are not manually tuned; they arise directly from the statistical structure of decision-tree paths and the multi-seed consistency check.

---

> ### Author Response · Authors · 2025-12-01
> **Response continued （3/4)**
>
> > Q5. Table 2 results are interesting and seems to suggest that strict adherence to the rules may not be necessary. How do you reconcile that with the central motivation of your paper?
>
> Thank you for the insightful observation. We agree that Table 2 shows an important phenomenon: **higher rule-compliance (RCR) does not necessarily translate into better utility**. This is consistent with our motivation, because **the goal of Component I is not to enforce rules as hard constraints, but to provide data-derived priors that guide the model toward realistic regions of the tabular space**.
>
> (1) Why high RCR does not imply higher utility.
> The prompt-only method achieves high RCR largely by producing repetitive patterns that happen to satisfy rule triggers. This “surface-level compliance’’ does not reflect genuine alignment with the true data distribution. In contrast, the rule-guided method yields RCR values closer to those of the real test data (Table 2) and exhibits significantly reduced distributional drift (Figure 3), indicating that it captures structure rather than memorizing rule fragments. Thus, RCR should not be maximized; it is most informative when its magnitude mirrors the target distribution.
>
> (2) Why this supports the paper’s motivation.
> Our rules are extracted from high-support, low-depth decision paths and are used as soft priors, not strict constraints. They help ensure generated samples lie within realistic support regions while still allowing the LLM to model nuances beyond the symbolic rules. The contrast with the DRL baseline, which achieves nearly 100% RCR but substantially worse downstream performance, further underscores that strict rule enforcement is neither necessary nor sufficient for good generative quality.
>
> In this sense, Table 2 reinforces our central motivation: **rule guidance should improve distributional fidelity, not maximize rule compliance**. The observed divergence between RCR and downstream utility is exactly what we expect when rules are used appropriately as soft, data-driven priors.
>
> ---
>
> >Q6. Splitting the report between the "seen" and unseen" data is very useful. It appears that the proposed method doesn't win in the "seen" data, but does well in the "unseen" data. Does this mean that when the LLMs have strong prior models of the data, trying to force this rule may be determental to the over all performance?
>
> Thank you for this thoughtful question. The seen/unseen split indeed reveals an interesting asymmetry, and we clarify how this relates to the role of rule guidance.
>
> (1) Seen datasets: small margins reflect a ceiling effect rather than degradation.
> On datasets such as Adult or Heart, LLMs already exhibit strong prior alignment with the tabular structure because these domains are well-represented in pretraining corpora. Consequently, the prompt-only baseline already produces samples with low distributional drift, leaving limited room for further improvement.
> Importantly, as shown in Table 1, ReFine does not harm performance on these datasets: the small differences lie well within the variance across random splits, and we observe no systematic degradation attributable to rule guidance.
>
> (2) Unseen datasets: gains appear precisely when prior alignment is weak.
> For domains that are unlikely to be covered in pretraining, LLM priors are clearly misaligned. Here, the extracted rules provide necessary structural anchors, allowing ReFine to correct distributional drift and substantially improve downstream performance. These gains hold across N∈{30,60,90,120,160,200,300,500}, indicating that rule guidance is particularly valuable for low-data regimes.
>
> (3) Does rule guidance become harmful when priors are strong?
> **Our results indicate no: ReFine behaves conservatively in prior-aligned settings because the rules used are weak, high-support, low-depth signals rather than strict constraints**. They nudge the model toward stable regions of support without overriding strong, accurate priors. Thus, the method is designed to be non-intrusive when the LLM already models the domain well, and corrective when it does not.
>
> In summary, the observed pattern supports the intended design: **ReFine is safe when LLM priors are strong and beneficial when priors are misaligned, which is a desirable behavior for practical deployment**.

---

> ### Author Response · Authors · 2025-12-01
> **Response continued （4/4)**
>
> >Q7.Can you give more details on the computation and fine-tuning costs?
>
> Thank you for the question. We are happy to clarify the computational footprint of ReFine, which is intentionally designed to avoid any LLM fine-tuning and to keep all components lightweight.
>
> (1) Cost of rule extraction and LLM-based consolidation.
> The tree models used for rule mining are extremely small (tens of samples per class), and training plus path extraction completes in under one second on a CPU for all datasets (N≤120). The subsequent rule generalization and denoising require only a handful of short LLM calls (<1k tokens each), which add negligible overhead relative to generation.
>
> (2) Cost of data generation.
> LLM generation uses standard prompting and is identical in cost to the prompt-only baseline; ReFine does not introduce any additional decoding steps or model modifications.
>
> (3) Cost of filtering.
> Proxy-density estimation and instance-level scoring scale linearly with the number of generated samples. The only relatively expensive step is the chunk-level scoring, where we train a small XGBoost model (30–120 samples) and record prediction trajectories across epochs; this step takes approximately 1–2 minutes on CPU. No GPUs are required.
>
> (4) No fine-tuning, minimal hyperparameters.
> ReFine introduces no LLM fine-tuning. Thresholds are determined from closed-form statistics, and the only tunable parameter (chunk size) is selected automatically via surprisal minimization. This keeps the method easy to deploy and highly reproducible.
>
> Overall, **the full main experimental suite (8 datasets × 4 data regimes × 10 seeds) runs comfortably on CPU-only environments**, in contrast to gradient-based tabular generators such as TabSyn or GREAT, which require GPU resources and substantial optimization.

---

### Official Review · Reviewer_PodE · 2025-11-01

**Soundness:** 2
**Presentation:** 2
**Contribution:** 2
**Rating:** 4
**Confidence:** 3

**Summary:**

The paper proposes a new framework called ReFine, which improves the generation of synthetic tabular data in settings where only a small amount of real data is available. The paper identify two main problems in existing LLM-based approaches: distributional drift (where generated samples deviate from real data distributions); and localized redundancy, (where generated data cluster excessively around common patterns). ReFine addresses these by combining two techniques: (i) Rule-Guided Generation, which extracts symbolic if–then rules from tree-based models to guide LLMs toward domain-consistent generation; and (ii) Dual-Granularity Filtering, which uses density-based filtering to remove redundant samples while preserving rare but informative ones.

**Strengths:**

The paper is well motivated.

While most synthetic data generation methods require abundant reference data, this work focuses on the low-data regime case, which is both important, underexplored and challenging.

The idea of combining symbolic rules (from decision trees) with LLM-based generative prompting to correct for distributional drift is interesting.

The paper provides the code implementing the method.

**Weaknesses:**

The paper’s experimental design and presentation could be strengthened in several key areas:

Results are reported only as averages, without showing variability across the 10 experimental runs, which is essential given the very small training subsamples used (30–120 samples). Reporting uncertainty measures would improve statistical validity.

The evaluation should also include additional criteria, especially data privacy, and explore larger datasets (up to 500–1000 samples) to better demonstrate the method’s generality. Comparisons against stronger baselines in the particular low data regime, such as TabPFN, would make the paper considerably stronger.

Finally, a few methodological sections (particularly 3.3.2 and 3.2) are difficult to follow and would benefit from clearer notation, illustrative examples, and expanded explanations (possibly in the Appendix).

Next, I describe in more detail each one of these issues.

Main issues:

Although line 309 states that the experiments were performed using 10 distinct random train/test splits, Table 1 only reports average F1 and R² scores. The paper should include measures of variability—such as standard deviations or, ideally, full score distributions—across these replications. Given that the experiments use very small training subsamples (sizes 30, 60, 90, and 120), substantial variability across runs is expected. Reporting uncertainty around the F1 and R² estimates is therefore essential for assessing the statistical reliability of the results.

The paper should also evaluate ReFine using additional synthetic data evaluation metrics. At a minimum, the paper should evaluate data privacy. (The data contamination issues might show up more clearly for the privacy metrics.)  Evaluating data fidelity would also make the paper stronger (but is probably not essential, since fidelity metrics tend to correlate well with machine learning efficiency).

I understand that the focus of the paper is on low data regime but, in my opinion, constraining the evaluations to training datasets containing 30 to 120 samples is a little too restrictive. (These ultra-low data regimes are not that common in practice.) I think the paper would be much more appealing if it could demonstrate the value of ReFine for datasets with up to 500 (or perhaps up to 1000) samples, which can still be quite challenging to other baseline generators, which usually require large training datasets for performing well (e.g., TabDDPM, TABSYN, GReaT, etc).

Also, the paper would benefit from the inclusion of other baselines, in particular, from comparisons against TabPFN [1, 2]. (Reference [2] describes how TabPFN can be used for synthetic data generation.) I think this is a particularly relevant baseline because it has been shown to achieve state-of-the-art performance in classification and regression tasks under low data regimes.  (Most of the current baselines evaluated in the paper are expected to perform poorly in the low data regime. Synthetic data generated by TabPFN might represent a stronger baseline in this setting.)

The paper provides no information regarding the tuning of the other baselines.

The paper could also benefit from improvements in the presentation. While the Introduction is well written and motivates well the proposed approach, the Methodology Section could be improved. For instance, Section 3.2 would benefit from an illustrative example describing how the association rules are extracted from the random forest trees, and how they are generalized, denoised and then converted to structured prompts for the LLM (such an illustrative example could be included in the Appendix for the sake of space).

I also found Section 3.3.2 difficult to follow. The paper needs to describe better the notation and elaborate more on the descriptions. For instance, what does $T$ represent in equation 2? I am guessing that $P_{M_t}(y_i | x_i) > 0.5 )$ represents the prediction of a classifier, but what about the regression task case where y_i is continuous? What do $\mu_{conf}$, $\mu_{uncert}$, $\sigma_{conf}$, and $\sigma_{uncert}$ represent in equation 4? (To be honest, I didn’t quite understand how the dual-granularity filtering is implemented. If explaining it in more detail will take space, I would suggest the paper add a detailed explanation in the Appendix, leaving only a high-level description in the main text.)

Additional comments/questions:

Why in Sections 4.3 and 4.4 the paper uses only 3 datasets? Presenting the analyses for all 8 datasets would make the results more compelling.

I enjoyed that the paper used dataset contamination tests for evaluating potential dataset contamination in the adopted benchmark datasets.

In line 239 the paper states “let DCR be the mixed-type distance from (Borisov et al., 2023)”. But what do you mean by mixed-type distance? Is it the way that Borisov et al handle categorical variables in their computation of DCR? If that is the case, it would be better to explicitly describe this procedure rather than referencing it indirectly.

In lines 96 to 98 the paper states that one of its main contributions was to identify the two key challenges to LLM-based synthetic data generation (i.e., distributional drift and localized redundancy). But, didn’t the references cited in the paragraphs starting in line 71 and ending in line 84 already pointed out these issues?

Typos:

Line 43: “which used” should be “which is used”

Line 112: “Many work” should be “Many works”

Line 122: “Layers Stoian et al. (2024b)” should be “Layers (Stoian et al., 2024b)”

Line 124: “Stoian & Giunchiglia (2025)” should be “(Stoian & Giunchiglia, 2025)”

Lines 237 to 238: “denote the N = …”  something is missing here

Line 238: “from (Borisov et al., 2023)” should be “from Borisov et al. (2023)”

References

[1] Hollmann et al. (2023). TabPFN: a transformer that solves small tabular classification problems in a second. ICLR 2023.

[2] Hollmann et al. (2025). Accurate predictions on small data with a tabular foundation model. Nature, 637, 319-326.



Initial recommendation:

Overall, I find it difficult to accurately assess the performance of ReFine relative to the baselines in its current form, and therefore I am inclined to recommend rejection at this stage. Nevertheless, the paper presents an interesting idea and addresses a meaningful and timely problem. I would be happy to revisit my assessment if the authors can clarify the issues outlined above and provide stronger empirical evidence supporting ReFine during the discussion period.

**Questions:**

See above.

---

> ### Author Response · Authors · 2025-12-01
> **Response to the reviewer's feedback**
>
> Thank you for your time and efforts in reviewing our paper. Please find our responses below to your concerns.
>
> >W1.Although line 309 states that the experiments were performed using 10 distinct random train/test splits, Table 1 only reports average F1 and R² scores. The paper should include measures of variability—such as standard deviations or, ideally, full score distributions—across these replications. Given that the experiments use very small training subsamples (sizes 30, 60, 90, and 120), substantial variability across runs is expected. Reporting uncertainty around the F1 and R² estimates is therefore essential for assessing the statistical reliability of the results.
>
> **A1**. We thank the reviewer for highlighting the importance of reporting uncertainty, especially under low‑data regimes where variance can indeed be substantial. Our experiments were conducted using 10 independent random splits for each dataset–N pair, as stated, and complete standard deviation statistics are available for all results.
>
> Due to space constraints and an oversight on our side, we only reported mean performance in Table 1. In this revised version, we include the full mean ± standard deviation results for the main comparisons (Table 10 & Table 11) across all datasets and baselines in **Appendix F**. These additions confirm that ReFine’s improvements remain consistent across splits, and the variance is well within expected ranges for low‑data tabular benchmarks.
>
> We appreciate the reviewer’s suggestion and agree that including these results strengthens the statistical reliability of our findings.
>
> ---
> >W2.The paper should also evaluate ReFine using additional synthetic data evaluation metrics. At a minimum, the paper should evaluate data privacy. (The data contamination issues might show up more clearly for the privacy metrics.) Evaluating data fidelity would also make the paper stronger (but is probably not essential, since fidelity metrics tend to correlate well with machine learning efficiency).
>
> **A2**. We thank the reviewer for raising the important point regarding privacy. We fully agree that synthetic tabular data generation methods should be assessed not only by downstream utility but also by privacy risk.
>
> 1.**Add Privacy Evaluation**: Following standard practices in the tabular data generation literature (e.g., GREAT), we evaluate ReFine’s privacy using the **Distance to Closest Record (DCR)** metric, which measures the distance between each synthetic sample and its nearest training sample.
> For each generated sample $s_{\text{syn}} \in D_{\text{syn}}$ and training sample $s_{\text{train}} \in D_{\text{train}}$, we compute:
>
> $\mathrm{DCR}(s_{\mathrm{syn}})=\min_{s_{\mathrm{train}}\in D_{\mathrm{train}}}\mathrm{Dist}(s_{\mathrm{train}},s_{\mathrm{syn}})$
>
>
>
> where Dist is the standard mixed-type L1 distance used in prior work (0/1 distance for categorical features and absolute difference for numerical features).
> This metric is widely used to detect record copying and potential LLM memorization.
>
> We now report full DCR distributions for four unseen datasets in **Appendix G**. The results show that ReFine's generated samples maintain a safe distance from training records (DCR values consistently > 0), with distributions closely resembling those between real test and train splits. This confirms that ReFine does not leak or memorize any original data—even when trained on as few as 30 records.
>
> 2.**Clarification on DCR usage in our method**:
> As DCR is also used in Section 3.3 for proxy density estimation, we clarify that this usage serves a different role:
> - In privacy evaluation, DCR assesses closeness to training data.
> - In dual-granularity filtering, we use DCR solely as a distance metric to assign synthetic samples to their nearest anchor for density estimation.
>
> This shared distance measure **does not introduce leakage**, since the filtering stage does not optimize DCR itself and never sees test data. We have updated Appendix G with a detailed clarification.
>
> We thank the reviewer again for highlighting this point and agree that the inclusion of DCR-based privacy analysis meaningfully strengthens the paper.

---

> ### Author Response · Authors · 2025-12-01
> **Response continued （1/4）**
>
> > W3. I understand that the focus of the paper is on low data regime but, in my opinion, constraining the evaluations to training datasets containing 30 to 120 samples is a little too restrictive. (These ultra-low data regimes are not that common in practice.) I think the paper would be much more appealing if it could demonstrate the value of ReFine for datasets with up to 500 (or perhaps up to 1000) samples, which can still be quite challenging to other baseline generators, which usually require large training datasets for performing well (e.g., TabDDPM, TABSYN, GReaT, etc).
>
>  > Also, the paper would benefit from the inclusion of other baselines, in particular, from comparisons against TabPFN [1, 2]. (Reference [2] describes how TabPFN can be used for synthetic data generation.) I think this is a particularly relevant baseline because it has been shown to achieve state-of-the-art performance in classification and regression tasks under low data regimes. (Most of the current baselines evaluated in the paper are expected to perform poorly in the low data regime. Synthetic data generated by TabPFN might represent a stronger baseline in this setting.)
>
> **A3**. We thank the reviewer for this constructive suggestion. While our work primarily targets ultra-low data regimes (e.g., n ≤ 120), which are common in privacy-constrained or scientific domains, we agree that evaluating in wider regimes enhances the paper's scope and applicability.
>
> 1.**Expanded training sizes (up to 500 samples)**:
> We have extended our experiments on four representative datasets (DISEASE, GAME, GPA, STUDENT) to include n={160, 200, 300, 500}. As shown in Table 12 (Appendix F):
> ***ReFine remains the best or second-best method across most settings, even when data size increases up to 500***.
> This demonstrates that our method generalizes well to moderate data regimes and continues to outperform baselines originally designed for large-scale training.
>
> 2.**Added TabPFN as generator and evaluator**:
> To address the reviewer’s request, we have additionally incorporated TabPFN into both the synthetic generation and evaluation pipelines:
>
> - We used the official open-source codebase [https://github.com/PriorLabs/TabPFN] and pretrained model weights [https://huggingface.co/Prior-Labs/tabpfn_2_5];
> - We fine-tuned the pretrained TabPFN model  by $D_{train}$  for synthetic generation following the protocol described in [1];
> - We further used TabPFN as the downstream evaluator, replacing XGBoost to test how well different generators support TabPFN-based tasks.
>
> The results are reported in **Table 12 and Table 13** (Appendix F). **ReFine remains  the best or second-best method in both TabPFN-generated and TabPFN-evaluated scenarios**.
>
> **Attention**：We note that using TabPFN both to generate and evaluate (on its own architecture) may introduce evaluation bias, but we still include these results for completeness, and provide average rank comparisons to facilitate fair analysis.
>
> Due to time constraints during the rebuttal period, we were not able to complete these extended experiments on all 8 datasets. However, we are currently running the full evaluation and commit to including results on all datasets in the next revision stage or camera-ready version.
>
> We appreciate the reviewer’s insightful suggestions, which have significantly improved the completeness of our evaluation. The additional experiments broaden the empirical scope and reinforce the core claims of ReFine.
> We hope these updates address your concerns and reflect the contribution and rigor of our work.
>
> Reference
>
> [1] Hollmann et al. (2025). Accurate predictions on small data with a tabular foundation model. Nature, 637, 319-326.

---

> > ### Author Response · Authors · 2025-12-01
> > **Response continued （2/4)**
> >
> > > W4. The paper provides no information regarding the tuning of the other baselines.
> >
> > > The paper could also benefit from improvements in the presentation. While the Introduction is well written and motivates well the proposed approach, the Methodology Section could be improved. For instance, Section 3.2 would benefit from an illustrative example describing how the association rules are extracted from the random forest trees, and how they are generalized, denoised and then converted to structured prompts for the LLM (such an illustrative example could be included in the Appendix for the sake of space).
> >
> > **A4**. We thank the reviewer for pointing out areas for clarification. We address each point below:
> >
> > 1. **Baseline tuning protocol**.
> > We provide detailed implementation, training, and hyperparameter configurations for all baselines (CTGAN, TABSYN, GReaT, DRL, etc.) in Appendix C. For each model, we closely followed the original papers or official implementations and tuned hyperparameters based on validation performance under each data regime. We will clarify this more explicitly in the main paper to avoid confusion.
> >
> > 2. **Illustrative example for rule extraction and prompt construction**.
> > We agree that a step-by-step example would improve understanding of the ReFine pipeline. As space was limited in the main text, we included the following components in the Appendix:
> >
> > - Appendix D.5 presents a concrete case study on the STUDENT dataset, showing how our method extracts association rules from a random forest and applies LLM-driven generalization and denoising.
> >
> > - Appendix H contains the actual prompt templates used for both Rule Generalization & Denoising stages, demonstrating how decision paths from random forests are converted into association rules.
> >
> > We will make cross-references to these appendices more prominent in Section 3.2 to guide readers more effectively.
> >
> > We appreciate the reviewer’s feedback and believe these clarifications will enhance the paper’s accessibility and reproducibility.

---

> ### Author Response · Authors · 2025-12-01
> **Response continued （3/4)**
>
> > W5. I also found Section 3.3.2 difficult to follow. The paper needs to describe better the notation and elaborate more on the descriptions. For instance, what does  represent in equation 2? I am guessing that  represents the prediction of a classifier, but what about the regression task case where y_i is continuous? What do , and  represent in equation 4? (To be honest, I didn’t quite understand how the dual-granularity filtering is implemented. If explaining it in more detail will take space, I would suggest the paper add a detailed explanation in the Appendix, leaving only a high-level description in the main text.)
>
> **A5**. We thank the reviewer for this helpful comment and agree that clearer notation will improve readability. We have made the following clarifications:
>
> (1) Meaning of \\(T\\) in Eq. (2):
> $T$ denotes the number of training iterations (epochs) of the reference model \\(M\\), during which we record its predictions on all synthetic samples.
> This follows the idea of modeling learning dynamics: samples on which the model’s predictions stabilize early exhibit lower uncertainty, while unstable predictions indicate noise or redundancy.
>
> (2) Regression tasks: For regression, we do not use a classification-based correctness indicator. Instead, for each sample $x_i$, we collect $T$ predictions $\\{\hat{y}\_{i,t}\\}\_{t=1}^{T}$ from $M$. We then compute a regression error, e.g., the RMSE over these predictions: $\\mathrm{RMSE}\_i = \\sqrt{\\frac{1}{T}\\sum\_{t=1}^{T} (\\hat{y}_{i,t} - y_i)^2}$, which serves as the quality signal for that sample.
>
>
> (3) Relation between classification/regression scores and $\mu_{\text{conf}},\ \sigma_{\text{conf}},\ \mu_{\text{uncert}},\ \sigma_{\text{uncert}}$
>
> For each synthetic sample $x_i$, we define two per-sample quantities:
>
> - **Classification.**
>   - $\mathrm{Conf}(x_i)$: the average predicted probability of the true class across the $T$ epochs,  $\mathrm{Conf}(x_i)=\frac{1}{T}\sum_{t=1}^{T}\mathbb{P}_{M_t}(y_i \mid x_i)$.
>   - $\mathrm{Uncert}(x_i)$: the standard deviation of these probabilities across epochs (predictive variability).
>
> - **Regression.**
>   - $\mathrm{Conf}(x_i)$: a monotone transformation of the regression error,  e.g., $\mathrm{Conf}(x_i)=-\mathrm{RMSE}_i$ or an equivalent rescaling, so that higher confidence corresponds to lower error.
>   - $\mathrm{Uncert}(x_i)$: the standard deviation of the predictions across epochs,  $\mathrm{Uncert}(x_i)=\sqrt{\frac{1}{T}\sum_{t=1}^{T}(\hat{y}_{i,t}-\bar{y}_i)^2}$,  where $\bar{y}_i$ is the mean prediction.
>
> The four quantities $\mu_{\text{conf}}, \sigma_{\text{conf}}, \mu_{\text{uncert}}, \sigma_{\text{uncert}}$ are then defined as the empirical mean and standard deviation of these per-sample scores over all synthetic samples:
>
> $$
> \mu\_{\mathrm{conf}}=\mathbb{E}[\mathrm{Conf}\(x)], \quad
> \sigma\_{\mathrm{conf}}=\mathrm{Std}[\mathrm{Conf}(x)],
> $$
>
> $$
> \mu_{\text{uncert}}=\mathbb{E}[\mathrm{Uncert}(x)], \quad
> \sigma_{\text{uncert}}=\mathrm{Std}[\mathrm{Uncert}(x)].
> $$
>
> They are used only to construct adaptive thresholds:
>
> $$
> \\mathrm{Conf}\_{\\text{thresh}}=\\mu_{\\text{conf}} - G(p)\\,\\sigma_{\\text{conf}},
> \\qquad
> \\mathrm{Uncert}\_{\\text{thresh}}=\\mu_{\\text{uncert}} + G(p)\\,\\sigma_{\\text{uncert}}.
> $$
>
> so that samples with unusually low confidence or unusually high uncertainty (relative to the dataset distribution) are filtered out.
>
> Due to space constraints, we kept the main text concise. As suggested, we will provide a clearer notation table, revised definitions, and full pseudocode explaining the dual‑granularity process in the the next revised version.
> We appreciate the reviewer’s careful reading and constructive suggestions.

---

> ### Author Response · Authors · 2025-12-01
> **Response continued （4/4)**
>
> >W6. In lines 96–98 the paper states that one of its main contributions was to identify two key challenges—distributional drift and localized redundancy—but didn’t the references in lines 71–84 already mention these issues?
>
> **A6**. We appreciate this insightful question.
>
> While prior works (e.g., CTAB-GAN+, TABSYN) have touched on related issues, such as mode collapse, data sparsity, or overfitting, they do not explicitly formulate the two distinct challenges we highlight: distributional drift (i.e., synthetic data deviating from the proxy distribution used for training) and localized redundancy (i.e., oversampling of dense regions with limited utility).
> *To the best of our knowledge, our work is the first to explicitly identify and systematically tackle both issues as key obstacles for LLM-based tabular generation in low-data regimes*.
>
> We agree that prior literature raises related concerns, and we will clarify this distinction in the camera-ready version by revising lines 71–98 to avoid overclaiming.
>
> >W7. Why in Sections 4.3 and 4.4 the paper uses only 3 datasets? Presenting the analyses for all 8 datasets would make the results more compelling.
>
> **A7**. We agree and appreciate this suggestion. Due to space and time constraints during the initial submission and rebuttal window, Sections 4.3 and 4.4 (distributional drift & redundancy analysis) focused on 3 representative datasets. We are actively extending this analysis to all 8 datasets and will include the results in the next revision or camera-ready version.
>
> **Additional**: We have also corrected the typos and unclear phrasing you helpfully pointed out. We sincerely thank the reviewer for the thorough reading, valuable feedback, and constructive suggestions that have helped strengthen both the clarity and technical depth of our submission.

---

### Note · Authors · 2026-01-15

**Comment:**

We have decided to withdraw this submission from ICLR.

After further consideration, we believe the current version would benefit from additional refinement and validation, and we prefer to pursue this work at a later stage.

We thank the reviewers and organizers for their time and effort

**Withdrawal Confirmation:**

I have read and agree with the venue's withdrawal policy on behalf of myself and my co-authors.